# Ergodic Generative Flows

**Leo Maxime Brunswic*** [1]  **Mateo Clemente*** [1]  **Rui Heng Yang** [1]  **Adam Sigal** [1]  **Amir Rasouli** [1]  **Yinchuan Li** [2]

## Abstract

Generative Flow Networks (GFNs) were initially introduced on directed acyclic graphs to sample from an unnormalized distribution density. Recent works have extended the theoretical framework for generative methods allowing more flexibility and enhancing application range. However, many challenges remain in training GFNs in continuous settings and for imitation learning (IL), including intractability of flow-matching loss, limited tests of non-acyclic training, and the need for a separate reward model in imitation learning. The present work proposes a family of generative flows called Ergodic Generative Flows (EGFs) which are used to address the aforementioned issues. First, we leverage ergodicity to build simple generative flows with finitely many globally defined transformations (diffeomorphisms) with universality guarantees and tractable flow-matching loss (FM loss). Second, we introduce a new loss involving cross-entropy coupled to weak flow-matching control, coined KL-weakFM loss. It is designed for IL training without a separate reward model. We evaluate IL-EGFs on toy 2D tasks and real-world datasets from NASA on the sphere, using the KL-weakFM loss. Additionally, we conduct toy 2D reinforcement learning experiments with a target reward using the FM loss.

## 1. Introduction

Generative models aim to sample from a target distribution $\kappa$ on a space $\mathcal{S}$, possibly conditioned on additional context or variables. The target distribution $\kappa$ is either specified by an unnormalized density with respect to a background measure $\lambda$, as in reinforcement learning (RL), or defined by a dataset of samples $\mathcal{D}$ as in imitation learning (IL). While RL (Brooks et al., 2011; Sutton, 2018; Haarnoja

*Equal contribution [1]Huawei Technologies Canada, Noah's Ark Laboratories [2]Huawei. Correspondence to: Leo Maxime Brunswic <leo.maxime.brunswic@h-partners.com>.

*Proceedings of the 42$^{nd}$ International Conference on Machine Learning*, Vancouver, Canada. PMLR 267, 2025. Copyright 2025 by the author(s).

et al., 2018; Bengio et al., 2023) and IL (Goodfellow et al., 2020; Ho et al., 2020; Song et al., 2020b; Rozen et al., 2021; Papamakarios et al., 2021) methods are often considered to be separate approaches, it is a common practice to first train a reward model from a dataset and then use RL techniques to solve an IL task (Chung et al., 2024; Zhang et al., 2022b;a; Sendera et al., 2024).

Among IL methods, normalizing flows (NFs), although capable of leveraging advanced neural ODEs (Chen et al., 2018; Grathwohl et al., 2018), have not achieved state-of-the-art performance due to architectural constraints (i.e. the requirement of sequentially combined parameterized diffeomorphisms) and numerical instabilities (Caterini et al., 2021; Verine et al., 2023).

While NFs are deterministic models, stochasticity has proven to be a crucial component of IL generative models, such as diffusion models as its simplest embodiment (Ho et al., 2020; Song et al., 2020b). However, diffusion models face a challenging trade-off between loss function tractability and the time-consuming denoising process for generation. This trade-off drives continued efforts to accelerate the diffusion inference process (Lu et al., 2022; Tang et al., 2025; Chen et al., 2024). Such issues are particularly important when the computational and time budgets of generation are highly constrained, which is typical on real-time tasks or mobile devices. The present work leverages the flexibility of Generative Flow Networks (GFNs) (Bengio et al., 2021; 2023) to address these issues by building simple yet highly expressive models with short sampling trajectories.

Generative Flow Networks (GFNs), originally formulated for RL tasks, are a family of generative methods designed to sample proportionally based on a reward function $r = \varphi_\kappa$, which represents the density of a target distribution $\kappa$. They were initially restricted to directed acyclic graphs, but have since been extended to other settings. Subsequent work has extended GFNs to generalize to continuous state spaces (Li et al., 2023; Lahlou et al., 2023) and non-acyclic structures (Brunswic et al., 2024).

Despite these advancements, GFNs still face four key challenges that hinder their application in RL and IL settings. First, although it has been suggested that GFNs themselves could be used as a reward model in the IL setting (see Bengio et al. (2023) p37), previous works (Zhang et al., 2022b;

Lahlou et al., 2023) train a separate reward model on samples from $\kappa$, effectively reducing the problem to an RL task. The need to train an additional reward model, which results in additional training and computation costs, stems from a lack of control of the so-called flow-matching (FM) property of the flow. Indeed, the FM-loss is zero by construction of the GFN self-defined reward model.

Second, the FM training loss presents significant challenges in continuous settings, as its evaluation is intractable in naive implementations (Li et al., 2023), necessitating the training of an imperfect estimator when directly enforcing the FM constraint. Addressing this issue requires either handcrafting or training an additional backward policy $\pi_{\leftarrow}^*$, or resorting to higher-variance loss functions, such as the Detailed Balance (DB) (Bengio et al., 2021) and Trajectory Balance (TB) (Malkin et al., 2022) losses.

Third, the acyclicity requirements (Lahlou et al., 2023) mean that additional structures must be manually crafted. Cycles appear naturally in naive implementations and RL environments. Brunswic et al. (2024) offers a theory for non-acyclic flows and stable FM losses, but these have yet to be tested in strongly non-acyclic settings.

Lastly, Li et al. (2023) argues that exact cycles are negligible in the continuous setting due to their zero probability, whereas Brunswic et al. (2024) argues that 0-flows, ergodic measures $\xi$ for $\pi_{\rightarrow}^*$, need to be addressed. Otherwise, the divergence-based losses used by Bengio et al. (2023) may become unstable in the presence of $\xi$ — a prediction that remains untested.

We present Ergodic Generative Flows (EGFs), which addresses the aforementioned key limitations. EGFs are built using a policy choosing at a given state from finitely many globally defined transformations, i.e. diffeomorphisms of the state space. This allows for tractable inverse flow policy, hence tractable FM-loss. Since we favor finitely many simple transformations, the universality of EGFs is non trivial. We provide provable guarantee with an *ergodicity* (Walters, 2000; Kifer, 2012) assumption of the group generated by the EGFs transformations.

**Main contributions**:

1. We extend the theoretical framework of generative flows, in particular providing quantitative versions of the sampling theorem for non-acylic generative flows.

2. We develop a theory of EGFs, presenting a universality criterion and demonstrating that, in any dimension on tori and spheres, this criterion can be fulfilled with four simple transformations.

3. We propose a coupled KL-weakFM loss to train EGFs directly for IL. This allows to train EGFs for IL without a separate reward model.

## 2. Preliminaries: GFN for RL and IL

In this work, we focus on generative flows in the continuous setting, encompassing scenarios where the state space $\mathcal{S}$ is either a vector space, $\mathcal{S} = \mathbb{R}^d$, or potentially a Riemannian manifold (Lee, 2018; Gemici et al., 2016). Both are instances of Polish[1] spaces that admit a natural finite background measure $\lambda$ (usually the Lebesgue, Haar (Halmos, 2013) or Riemmanian volume measures).

A measure $\mu$ is dominated by $\nu$, denoted $\mu \ll \nu$ if $\nu(A) = 0 \Rightarrow \mu(A) = 0$ for any measurable $A \subset \mathcal{S}$; then, the Radon-Nikodym derivative of $\mu$ with respect to $\nu$, denoted $\frac{d\mu}{d\nu} \in L^1(\nu)$, is the unique $\nu$-integrable function $\varphi$ such that $\varphi\nu = \mu$. A Markov kernel $\pi : \mathcal{X} \rightarrow \mathcal{Y}$ is a stochastic map denoted $\pi(x)$. The image measure of $\mu$ by $\pi$ denoted $\mu\pi$ is defined by $\mu\pi(A) = \int_{x \in \mathcal{X}} \mathbb{P}(\pi(x) \in A)d\mu(x)$ for any measurable $A \subset \mathcal{Y}$. Their tensor product is the kernel $\mu \otimes \pi(A \rightarrow B) := \int_{x \in A} \mathbb{P}(\pi(x) \in B)d\mu(x)$. Even when $\mu$ is an unnormalized distribution, we write $x \sim \mu$ for "the law of $x$ is $\frac{1}{\mu(\mathcal{S})}\mu$".

We adopt a formulation of generative flows that differs from Lahlou et al. (2023) and Brunswic et al. (2024), providing the necessary flexibility for the subsequent sections. Let $(\mathcal{S}, \lambda)$ be a Polish space with a finite background measure. A generative flow on $\mathcal{S}$ is the data of a star forward policy, i.e. a Markov kernel $\pi_{\rightarrow}^* : \mathcal{S} \rightarrow \mathcal{S}$, and a star outflow, i.e. a finite non-negative measure $F_{\rightarrow}^*$ on $\mathcal{S}$.

With unnormalized initial and terminal distributions $F_{\text{init}}$ and $F_{\text{term}}$, it induces a Markov chain $(\underline{s}_t)_{g \geq 1}$ defined by $\underline{s}_1 \sim F_{\text{init}}$ and $\underline{s}_{t+1} = \pi_{\rightarrow}^*(\underline{s}_t)$. Define the sampling time $\tau \in \mathbb{N}_{\geq 1}$ as the random variable given by[2] $\mathbb{P}(\tau \geq 1) = 1$ and $\mathbb{P}(\tau = t | \tau \geq t) = \frac{dF_{\text{term}}}{d(F_{\text{term}} + F_{\rightarrow}^*)}(\underline{s}_t)$. The sampler associated with a generative flow is obtained by emulating the Markov chain $(\underline{s}_t)_{t \geq 1}$ and sampling $\underline{s}_\tau$. By extending the sampling Markov chain with a source $s_0$ and sink $s_f$ state, we can summarize the flow as follows:

$$s_0 \xrightarrow{F_{\text{init}}} \mathcal{S} \xrightarrow{F_{\text{term}}} s_f \ . \tag{1}$$

The following theorem provides theoretical guarantees on the sampler associated with a generative flow.

**Theorem 2.1** ((Bengio et al., 2021; Brunswic et al., 2024))**.** *Let $F_{\text{init}}$, and $F_{\text{term}}$ be unnormalized distributions with $F_{\text{term}} \neq 0$ and let $(\pi_{\rightarrow}^*, F_{\rightarrow}^*)$ be a generative flow. If the flow-matching constraint*

$$F_{\text{init}} + F_{\rightarrow}^* \pi_{\rightarrow}^* = F_{\text{term}} + F_{\rightarrow}^* \tag{2}$$

---

[1] A topological measurable space is Polish if its topology is completely metrizable and it is endowed with its Borel $\sigma$-algebra.

[2] Clearly $F_{\text{term}} \ll F_{\text{term}} + F_{\rightarrow}^*$ so the Radon-Nikodym derivative is well-defined.

*is satisfied then $F_{\text{init}}(\mathcal{S}) = F_{\text{term}}(\mathcal{S})$ and:*

$$\mathbb{E}(\tau) \leq \frac{F_{\rightarrow}^*(\mathcal{S})}{F_{\text{init}}(\mathcal{S})} + 1, \qquad \underline{s}_\tau \sim F_{\text{term}}.$$

The pair $(F_{\rightarrow}^*, \pi_{\rightarrow}^*)$ is trained using gradient descent to minimize a loss function that enforces the flow-matching constraint in equation (2). Once convergence is achieved, the generative flow sampler yields samples of the random variable $s_\tau$. Hence, by Theorem 2.1, the generative flow sampler samples from $F_{\text{term}}$. For example, we can use the most straightforward stable FM-loss:

$$\mathcal{L}_{\text{FM},q}^{\text{stable}} = \mathbb{E}_{s \sim \nu_{\text{train}}} \left[ (f_{\text{init}} + f_{\leftarrow}^* - f_{\text{term}} - f_{\rightarrow}^*)^q (s) \right] \quad (3)$$

where $f_{\text{init}} = \frac{dF_{\text{init}}}{d\lambda}$, $f_{\text{term}} = \frac{dF_{\text{term}}}{d\lambda}$ and $f_{\leftarrow}^* = \frac{dF_{\leftarrow}^*}{d\lambda} = \frac{d(F_{\rightarrow}^*,\pi_{\rightarrow}^*)}{d\lambda}$.

In general, both $f_{\text{init}}$ and $f_{\rightarrow}^*$ are determined by the chosen parametrization, with $f_{\rightarrow}^*$ being problem-dependent, while $f_{\leftarrow}^*$ requires the computation of an image measure density. However, the previously mentioned methods outlined above face several limitations:

**Limitation ①:** The first limitation arises in IL. $f_{\text{term}}$ is unknown so one has to build a density model for $F_{\text{term}}$. Disregarding solutions based on separate energy models, we can attempt to set $F_{\text{term}} = F_{\text{init}} + F_{\leftarrow}^* - F_{\rightarrow}^*$, assuming $F_{\leftarrow}^*$ is tractable. We then use a cross-entropy loss to train $F_{\text{term}}$ to approximate $\kappa$. Despite the FM loss formally and trivially becoming 0, $F_{\text{term}}$ may not be positive. Therefore, the existing theoretical framework fails and so does the sampling. At this stage, we move beyond the existing theoretical framework.

**Limitation ②:** The second limitation arises in RL. At the inference time, the reward may be unknown, either because of its costly nature or because it is simply inaccessible. A straightforward solution consists of using $F_{\text{term}} = \kappa$ during training and $F_{\text{term}} = F_{\text{init}} + F_{\leftarrow}^* - F_{\rightarrow}^*$ during inference. However, this reward-less sampling is ill-controlled because $F_{\text{term}}$ may not be positive.

**Limitation ③:** Both training and inference are based on the tractability of $f_{\leftarrow}^*$. For instance, naive forward policies, such as those used in diffusion models add noise in $\mathbb{R}^d$. These policies are typically defined as $\pi_{\rightarrow}^*(x) = m(x) + \epsilon$, where $m$ is a deterministic model and $\epsilon \sim \mathcal{N}(0, \eta)$ represents Gaussian noise. This leads to an intractable star inflow as the integral in equation (4) becomes computationally intensive.

$$f_{\leftarrow}^*(y) \propto \int_{x \in \mathbb{R}^d} f_{\rightarrow}^*(x) \exp\left(-\frac{(m(x)-y)^2}{2\eta^2}\right) d\lambda(x). \tag{4}$$

**Limitation ④:** Although Li et al. (2023) argues that exact cycles are negligible in the continuous setting due to their

zero probability, Brunswic et al. (2024) disputes that their generalization called 0-flows must still be addressed. 0-flows are measures $\xi$ for which $\pi_{\rightarrow}^*$ is ergodic (Walters, 2000). Brunswic et al. (2024) predicts that divergence-based losses used by Bengio et al. (2023) are unstable if such a $\xi$ exists. This has yet to be tested.

## 3. Ergodic Generative Flows for RL and IL

To address the four issues outlined above, we propose Ergodic Generative Flows (EGFs).

**Definition 3.1** (Ergodic Generative Flows)**.** Let $\mathcal{S}$ be Riemannian manifold endowed with its volume measure $\lambda$ and let $(\Phi_i)_{i=1}^p$ be a family of diffeomorphisms $\mathcal{S} \rightarrow \mathcal{S}$. An EGF is a generative flow, where

$$\pi_{\rightarrow}^*(s) = \sum_{i=1}^p \alpha_{\rightarrow}^i(s) \delta_{\Phi_i(s)} \tag{5}$$

for some policy $\alpha_{\rightarrow} : \mathcal{S} \rightarrow [0,1]^p$, and such that the group of diffeomorphisms generated by the $\Phi_i$ is topologically ergodic. That is for all $x, y \in \mathcal{S}$ and any neighborhood $\mathcal{U}$ of $y$, there exists a sequence $i_1, \cdots, i_t$ such that $x\Phi_{i_1}\Phi_{i_2}\cdots\Phi_{i_t} \in \mathcal{U}$.

There are two key ideas that motivate this definition:

**Tractability** of star inflow $f_{\leftarrow}^*$ is achieved by only using finitely many diffeomorphisms. Let say that $\mathcal{S} = \mathbb{R}^d$ endowed with Lebesgue measure $\lambda$ and that each move $\Phi_i$ is a parametrizable diffeomorphism. Then, a closed form formula for the backward policy may be deduced from a detailed balance argument, see appendix A for details:

$$\pi_{\leftarrow}^*(s) = \sum_{i=1}^p \alpha_{\leftarrow}^i(s) \delta_{\Phi_i^{-1}(s)} \tag{6}$$

$$\alpha_{\leftarrow}^i(s) = \frac{(\alpha_{\rightarrow}^i f_{\rightarrow}^*) \circ \Phi_i^{-1}(s)|J_s\Phi_i^{-1}|}{\sum_j (\alpha_{\rightarrow}^j f_{\rightarrow}^*) \circ \Phi_j^{-1}(s)|J_s\Phi_j^{-1}|} \tag{7}$$

where $J_s$ denotes the Jacobian matrix at $s$. Therefore, we have a closed-form formula for the density of the star inflow,

$$f_{\leftarrow}^*(s) = \sum_{i=1}^p (\alpha_{\rightarrow}^i f_{\rightarrow}^*) \circ \Phi_i^{-1}(s)|\det J_s\Phi_i^{-1}|. \tag{8}$$

All terms in equation (8) are tractable, so $f_{\leftarrow}^*$ is tractable as long as the number of diffeomorphisms $p$ is small. As a consequence, the flow-matching loss given in Equation (3) is tractable for $p$ small enough.

**Expressivity** can potentially be an issue if the number of diffeomorphisms is small and each move is too simple (Rezende et al., 2020). Ergodicity guarantees that the parameterized family of EGFs is able to be flow-matching for

any non-zero $F_{\text{init}} \ll \lambda$ and $F_{\text{term}} \ll \lambda$ even with simple transformations, such as affine maps on tori and rotations on spheres.

The following sub-sections are organized as follow: Section 3.1 establishes the universality of EGFs. In Section 3.2, we present theoretical advancements for general generative flows, which forms the basis for the design of the KL-weakFM loss used IL training, described in Section 3.3.

## 3.1. Universality From Ergodicity

We begin with a formalization of universality in conjunction with the master universality Theorem.

**Definition 3.2.** On a state space $(\mathcal{S}, \lambda)$, a family of generative flow family is universal if for any two distributions $\mu, \nu \ll \lambda$ with bounded density there is a flow $(f_{\rightarrow}^*, \pi_{\rightarrow}^*)$ in the family that is flow matching for $F_{\text{init}} = \nu$ and $F_{\text{term}} = \mu$.

We begin with the presentation of our master universality theorem in section 3.1.1. This theorem is then applied to tori and spheres, providing simple examples of universal families, as detailed in sections 3.1.2 and 3.1.3.

### 3.1.1. MASTER UNIVERSALITY THEOREM

The next theorem states that if the family contains a sufficiently strong ergodic policy $\pi_{\rightarrow}^*$, then the family is universal. More precisely,

**Definition 3.3.** A Markov kernel $\pi$ on $(\mathcal{S}, \lambda)$ is summably $L^2$-mixing if $\pi$ admits an invariant measure $\widehat{\lambda}$ equivalent to $\lambda$ and such that the $L^2$-mixing coefficients given by,

$$\gamma_n := \sup_{\varphi \in H} \int_{s \in \mathcal{S}} (\varphi \pi^n(s))^2 \, d\widehat{\lambda}(s), \qquad (9)$$

with $H = \{\varphi \in L^2(\lambda) \mid \|\varphi\|_2 = 1 \text{ and } \int_{s \in \mathcal{S}} \varphi(s) = 0\}$ are summable: $\sum_{n \geq 0} \gamma_n < +\infty$.

The technical summably $L^2$-mixing property informally means that iterating the Markov kernel $\pi$ is averaging out functions fast enough to ensure that the sum of errors is finite. In contrast, ergodicity alone ensures the convergence of $\varphi \pi^n$ in the much weaker Cesaro sense.

This property is in particular satisfied if the policy $\pi$ induces a Koopman operator $\phi \mapsto \frac{d(\phi\lambda)\pi}{d\lambda}$ on $L^2(\mathcal{S})$, which has a so-called *spectral gap* (Conze & Guivarc'H, 2013). Informally the spectrum of the Koopman operator is bounded away from 1 on the subspace $\{\phi \mid \int \phi d\lambda = 0\}$. In this case, the convergence of iterates by $\pi$ is exponential.

**Theorem 3.4.** *A parameterized family of EGF is universal provided that it contains some summably $L^2$-mixing $\pi_{\rightarrow}^*$ and that the set of $f_{\rightarrow}^*$ is dense in $L^2(\mathcal{S}, \lambda)$.*

### 3.1.2. ERGODICITY ON TORI

For the sake of simplicity, let $\mathbb{T}^d$ be a flat torus of side 1 endowed with Lebesgue measure $\lambda$ and let $P : \mathbb{R}^d \to \mathbb{T}^d$ be the natural universal covering projection. The simplest family of transformations is the group of affine transformations,

$$\text{Aff}(\mathbb{T}^d) := \{x \mapsto Ax + b \mid b \in \mathbb{R}^d \text{ and } A \in \text{SL}_d(\mathbb{Z})\}, \qquad (10)$$

with $\text{SL}_d(\mathbb{Z})$ denoting the set of square matrices of order $d$ with integral coefficients and determinant 1.

Given that $(\Phi_i)_{i=1}^p$ are all in $\text{Aff}(\mathbb{R}^d)$, we can build a continuous family of generative flows using equation (5), which is then parameterized by the translation part of each $\Phi_i$ together with a model having a softmax head $\alpha_{\rightarrow}$ and a scalar head $f_{\rightarrow}^*$. We call such a family an *affine toroidal* family.

**Theorem 3.5.** *Any affine toroidal family is a universal EGF family provided mild technical assumptions on the $(\Phi_i)_{i=1}^p$.*

The "technical assumptions" above is in particular satisfied if the group generated by the linear part of the $\Phi_i$ is the whole group $\text{SL}_d(\mathbb{Z})$. It is shown in Conder et al. (2025) that $\text{SL}_d(\mathbb{Z})$ has a presentation with two generators. Therefore, in any dimension, EGF are universal with $p = 4$ and well-chosen $(\Phi_i)_{i=1}^4$: the two generators and their inverse.

### 3.1.3. ERGODICITY ON SPHERES

Consider the round sphere $\mathbb{S}^d = \{x \in \mathbb{R}^{d+1} \mid \|x\| = 1\}$ endowed with its natural Riemannian volume measure $\lambda$. Again a simple family of diffeomorphisms is provided by the so-called projective transformations,

$$\text{PGL}_{d+1}(\mathbb{R}) := \{x \mapsto Ax/\|Ax\| \mid A \in \text{GL}_{d+1}(\mathbb{R})\}, \qquad (11)$$

where $\text{GL}_{d+1}(\mathbb{R})$ is the group of invertible matrices of order $d+1$. Similar to tori, we build a continuous family of EGFs using equation (5) by parametrizing $(\Phi_i)_{i=1}^p$ in $\text{PGL}_{d+1}(\mathbb{R})$ which is a Lie group, and choose a tractable model that have a softmax head $\alpha_{\rightarrow}$ and a scalar head $f_{\rightarrow}^*$. Such a family is called a *projective spherical* family. We focus on the sub-family of *isometry spherical* family composed of rotations $\Phi_i \in \text{SO}_{d+1}(\mathbb{R})$ for which we have theoretical guarantees.

**Theorem 3.6.** *An isometry spherical family is a universal EGF family, provided technical assumptions on the $(\Phi_i)_{i=1}^p$.*

The technical assumption is satisfied if the $\Phi_i$ have algebraic coefficients and are frozen: this is a consequence of the main theorem of Bourgain & Gamburd (2012) (see also Benoist & de Saxcé (2016)) that guarantees a spectral gap under this assumption. Furthermore, the main Theorem of Breuillard & Gelander (2003) allows to build a dense subgroup of $\text{SO}_d(\mathbb{R})$ with two generators and algebraic coefficients. Therefore, EGF are universal on spheres with

$p = 4$ and well-chosen $(\Phi_i)_{i=1}^4$: the two generators and their inverse.

## 3.2. Quantitative Sampling Theorem

We present a quantitative sampling theorem for generative flows. The key idea of the proof is that any generative flow matches the flow up to a transformation of the initial and terminal distributions.

**Definition 3.7** (Virtual initial and terminal flow). Let $(\pi_\rightarrow^*, f_\rightarrow^*)$ be a generative flow and let $F_{\text{init}}, F_{\text{term}}$ be initial and terminal distributions. Define the initial and terminal errors as $\delta F_{\text{init}} := (F_{\text{init}} + F_\leftarrow^* - F_{\text{term}} - F_\rightarrow^*)^-$ and $\delta F_{\text{term}} := (F_{\text{init}} + F_\leftarrow^* - F_{\text{term}} - F_\rightarrow^*)^+$ respectively. Define the positive and negative parts of a measure $\mu$ as $\mu^\pm$. The virtual initial and terminal distributions are $\widehat{F}_{\text{init}} := F_{\text{init}} + \delta F_{\text{init}}$ and $\widehat{F}_{\text{term}} := F_{\text{term}} + \delta F_{\text{term}}$.

We note that once the initial and terminal distributions $F_{\text{init}}, F_{\text{term}}$ are specified, then any generative flow $(\pi_\rightarrow^*, f_\rightarrow^*)$ is a FM with respect to the virtual initial and terminal distributions, $\widehat{F}_{\text{init}}$ and $\widehat{F}_{\text{term}}$. This enables the presentation of a quantitative formulation of the sampling theorem for generative flows.

**Theorem 3.8** (Quantitative Sampling of Generative Flows). *Let $(\pi_\rightarrow^*, f_\rightarrow^*)$ be a generative flow and let $F_{\text{init}}$ and $F_{\text{term}}$ initial and terminal distributions. Assume that $F_{\text{init}} \neq 0$ and consider the sample $s_\tau$ of the generative flow Markov chain from $F_{\text{init}}$ to $\widehat{F}_{\text{term}}$ then:*

$$\text{TV}\left(s_\tau \,\middle\|\, \frac{1}{\widehat{F}_{\text{term}}(\mathcal{S})} \widehat{F}_{\text{term}}\right) \leq \frac{\delta F_{\text{init}}(\mathcal{S})}{\widehat{F}_{\text{term}}(\mathcal{S})}, \quad (12)$$

*with* TV *the total variation.*

**Corollary 3.9.** *For a generative flow trained on the target $\kappa$ with $F_{\text{init}}(\mathcal{S}) = 1$ and using $F_{\text{term}} = \widehat{\kappa} := (F_{\text{init}} + F_\leftarrow^* - F_\rightarrow^*)^+$ during inference, the sampling error is controlled by:*

$$\text{TV}\left(s_\tau \,\middle\|\, \frac{\kappa}{\kappa(\mathcal{S})}\right) \leq \frac{\delta}{1+\delta} + \text{TV}\left(\frac{\widehat{\kappa}}{\widehat{\kappa}(\mathcal{S})} \,\middle\|\, \frac{\kappa}{\kappa(\mathcal{S})}\right), \quad (13)$$

*with $\delta := (F_{\text{init}} + F_\leftarrow^* - F_\rightarrow^*)^- (\mathcal{S})$.*

Commonly the training loss is designed to control the total variation term in the upper bound. However, a secondary regularization term controlling $\delta = \delta F_{\text{init}}$ may be added to enhance reward-less sampling.

To substantiate this quantitative bound, let's consider the RL case with a reward $r$ defined on a finite DAG $(\mathcal{V}, \mathcal{E})$ with $\mathcal{V} = \mathcal{S} \cup \{s_0, s_f\}$, trained using a flow-matching loss, say $\mathcal{L}_{\text{FM},q}^{\text{stable}}$. Let's take $\nu_{\text{train}}$ the uniform distribution on the vertices and $q = 1$, then $\mathcal{L}_{\text{FM},q}^{\text{stable}} = \frac{1}{|\mathcal{V}|}\left(\delta + \sum_{s \in \mathcal{S}} |\widehat{r}(s) - r(s)|\right)$ with $\widehat{r}(s)$ the density of $\widehat{\kappa}$

with respect to the counting measure. We may also rewrite $\text{TV}\left(\frac{\widehat{\kappa}}{\widehat{\kappa}(\mathcal{S})} \,\middle\|\, \frac{\kappa}{\kappa(\mathcal{S})}\right) = \frac{1}{2} \sum_{s \in \mathcal{S}} |\frac{\widehat{r}(s)}{\sum_{s' \in \mathcal{S}} \widehat{r}(s')} - \frac{r(s)}{\sum_{s' \in \mathcal{S}} r(s')}|$ so that if $Z = \sum_{s \in \mathcal{S}} r(s)$ then

$$\text{TV}\left(s_\tau \,\middle\|\, \frac{\kappa}{\kappa(\mathcal{S})}\right) \leq \left(1 + \frac{1}{Z}\right) |\mathcal{V}| \mathcal{L}_{\text{FM},q}^{\text{stable}}. \quad (14)$$

## 3.3. KL-WeakFM loss and IL algorithm

Henceforth, we assume that $\mathcal{S}$ is a Polish space equipped with a background measure $\lambda$ and $\kappa$ is a target probability distribution.

Leveraging Theorem 3.8, we design the KL-weakFM loss, $\mathcal{L}_{KL-wFM}$, which enables IL training of a generative flow without requiring a separate reward model. Let $\delta f_{\text{init}} = \min(0, f_{\text{init}} + f_\leftarrow^* - f_\rightarrow^*)$ and $\widehat{f}_{\text{term}} = \max(0, f_{\text{init}} + f_\leftarrow^* - f_\rightarrow^*)$. The KL-weakFM loss can then be defined as follows:

$$\mathcal{L}_{KL-wFM}(\theta) :=$$
$$b\mathbb{E}_{s \sim \nu_{\text{train}}} \delta f_{\text{init}}(s) - \mathbb{E}_{s \sim \kappa} \log \widehat{f}_{\text{term}}(s), \quad (15)$$

for some training distribution of paths $\nu_{\text{train}}$ and $b > 0$.

The name of the loss is derived from two components: on the one hand, the cross-entropy term, which controls the Kullback-Leibler divergence between $\widehat{F}_{\text{term}}$ and $\kappa$; on the other hand, the term $\mathbb{E}_{s \sim \nu_{\text{train}}} \delta f_{\text{init}}(s)$, which resembles the FM-loss but controls only the negative part of the FM defect. We now provide a detailed description and motivation for $\mathcal{L}_{KL-wFM}$. The discussion begins with a corollary of Theorem 3.8, based on Pinsker's inequality:

**Corollary 3.10.** *Let $(\pi_\rightarrow^*, f_\rightarrow^*)$ be a generative flow. We train the generative flow on target probability distribution $\kappa$ with $F_{\text{init}}(\mathcal{S}) = 1$. By using $F_{\text{term}} = \widehat{\kappa} := (F_{\text{init}} + F_\leftarrow^* - F_\rightarrow^*)^+$ during inference, the sampling error is bounded as follow:*

$$\text{TV}(s_\tau \| \kappa) \leq \frac{\delta}{1+\delta} + \sqrt{\frac{1}{2}\text{KL}\left(\frac{\widehat{\kappa}}{\widehat{\kappa}(\mathcal{S})} \,\middle\|\, \kappa\right)}, (16)$$

*with $\delta := \int_{s \in \mathcal{S}} \delta f_{\text{init}}(s) d\lambda(s)$.*

First, the weak-FM term $\mathbb{E}_s \sum_{t=1}^\tau (\delta f_{\text{init}}(\underline{s}_t))^2$ in equation (15) controls the term $\frac{\delta}{1+\delta}$ in equation (16).

Second, since the target $\kappa$ has unknown density, we employ reward-less inference, using $F_{\text{term}} = (F_{\text{init}} + F_\leftarrow^* - F_\rightarrow^*)^+$. It is therefore natural to directly train the density of $(F_{\text{init}} + F_\leftarrow^* - F_\rightarrow^*)^+$ to match $\kappa$ using cross-entropy, leading to the second term on the right-hand side of Equation (15). Corollary 3.10 demonstrates that controlling the Kullback-Leibler divergence through cross-entropy also controls the total variation sampling error.

**Algorithm 1** Ergodic flow: IL training

---

**Input:** A list of trainable bi-Lipchitz maps $\Phi_1, \cdots, \Phi_p$
**Input:** A softmax model $\alpha : \mathcal{S} \to \mathcal{P}(\{1, \cdots, p\})$
**Input:** A trainable star outflow model $f_\to^* : \mathcal{S} \to \mathbb{R}_+$
**Input:** A target samplable distribution $\kappa$
**Input:** A source $F_{\text{init}}$ samplable and of density $f_{\text{init}}$
**repeat**

Fill replay buffer $\mathcal{B} = \mathcal{B}_\to \cup \mathcal{B}_\leftarrow$ with $B$ trajectories $(\underline{s}_t)_{t=1}^{t_{\max}}$ using $(F_{\text{init}}, \pi_\to^*)$ for $\mathcal{B}_\to$ and $(\kappa, \pi_\leftarrow^*)$ for $\mathcal{B}_\leftarrow$ with $t \in \{1, \cdots, \tau\}$
Minimization step of

$$\mathcal{L} = \frac{1}{2B} \sum_{t=1}^{\tau} \delta f_{\text{init}}(\underline{s}_t) p_t(\underline{s}) - b\mathbb{E}_{s\sim\kappa} \log \widehat{f}_{\text{term}}(s),$$

with

$$
\begin{aligned}
p_t(\underline{s}) &= \prod_{t'<t} \left(1 + (\widehat{f}_{\text{term}}/f_\to^*)(\underline{s}_{t'})\right)^{-1} &&\text{if } \underline{s} \in \mathcal{B}_\to \\
&= \prod_{t'<t} (1 + (\delta f_{\text{init}}/f_\leftarrow^*)(\underline{s}_{t'}))^{-1} &&\text{if } \underline{s} \in \mathcal{B}_\leftarrow
\end{aligned}
$$

**until** converged

---

Third, more precisely, the KL term may be decomposed into

$$
\begin{aligned}
&\text{KL}\left(\frac{\widehat{\kappa}}{\widehat{\kappa}(\mathcal{S})} \middle\| \kappa\right) = \\
&\underbrace{-\mathbb{E}_{s\sim\kappa} \log \widehat{f}_{\text{term}}(s)}_{\text{cross-entropy}} + \underbrace{\log \int_{s\in\mathcal{S}} \widehat{f}_{\text{term}}(s) d\lambda(s)}_{\text{normalization}} - \mathcal{H}(\kappa),
\end{aligned}
$$

$$(17)$$

where $\mathcal{H}(\kappa) = -\mathbb{E}_{s\sim\kappa} \log \frac{d\kappa}{d\lambda}(s)$ is the entropy of $\kappa$. Since the normalization constant is unknown a priori, the cross-entropy is insufficient to fully control the KL divergence. However,

$$\int_{s\in\mathcal{S}} \widehat{f}_{\text{term}}(s) d\lambda(s) = 1 + \int_{s\in\mathcal{S}} \delta f_{\text{init}}(s) d\lambda(s). \quad (18)$$

Therefore, the weak-FM term in equation (15) also controls the normalization factor of $\widehat{f}_{\text{term}}$.

# 4. Experiments

We proceed with experiments wherein the state space $\mathcal{S}$ is either a flat torus $\mathbb{T}^2$, or sphere $\mathbb{S}^2$. An EGF family is built by randomly choosing diffeomorphisms from affine toroidal families on $\mathbb{T}^2$ and from isometry spherical families on $\mathbb{S}^2$, together with two multi-layer perceptron (MLP) models: one for $f_\to^*$ and one for $\alpha_\to$. The MLPs are $\tanh$ hyperbolic activated and initialized using orthogonal initialization

(Saxe et al., 2013; Hu et al., 2020). We use the AdamW optimizer (Kingma & Ba, 2015; Loshchilov & Hutter, 2019) for training.

The KL-WeakFM loss tends to favor $\widehat{F}_{\text{term}}$ positive on the whole $\mathcal{S}$, therefore, even with highly localized target distributions, the EGF reward model tends to have a small background inducing unwanted outliers. We filter it out by replacing $\widehat{f}_{\text{term}}$ with $\mathbf{1}_{\widehat{f}_{\text{term}}>\eta} \widehat{f}_{\text{term}}$ during inference, where $\eta$ is chosen to minimize negative likelihood on a validation dataset. See Appendix D.1 for details.

## 4.1. RL Experiments

Two dimensional distributions are tested in RL settings, which allow us to conduct sanity-check on the tractability of the EGF, its stability and expressivity. An EGF on $\mathcal{S} = \mathbb{T}^2$ is built with 16 transformations (8 translations and 2 elements of $\text{SL}_d(\mathbb{Z})$ together with their inverse). Their MLPs have 5 hidden layers of width 32 to parameterize $f_\to^*$ and $\pi_\to^*$. This EGF is trained either with stable $\mathcal{L}_{\text{FM}}^{\text{stable}}$ or unstable $\mathcal{L}_{\text{FM}}^{\text{div}}$ FM-losses, with or without regularization $\mathcal{R}$ (see equations (3),(19) and (20)).

$$\mathcal{L}_{\text{FM}}^{\text{div}} = \mathbb{E}_{\underline{s}} \sum_{t=1}^{\tau} \log\left(\frac{f_\leftarrow^* + f_{\text{init}}}{f_\to^* + f_{\text{term}}}\right)^2 (\underline{s}_t) \quad (19)$$

$$\mathcal{R} = \mathbb{E}_{\underline{s}} \sum_{t=1}^{\tau} (f_\to^*)^2 (\underline{s}_t). \quad (20)$$

The regularization $\mathcal{R}$ is motivated by the stability Theory of Brunswic et al. (2024): as long as the directional derivative of $\mathcal{R}$ is positive on 0-flows directions, such a regularization helps reducing the unstability of a loss. See also Morozov et al. (2025) for a study of the impact of such regularizations for detailed balance losses on graphs.

A stability comparison to divergence-based FM loss (see figure 1) shows the expressivity of very small EGF as well as the unstability $\mathcal{L}_{\text{FM}}^{\text{div}}$ as predicted by Brunswic et al. (2024)(refer to limitation 4 in section 2).

## 4.2. IL Experiments

We proceed with imitation learning experiments on tori $\mathbb{T}^2$ and spheres $\mathbb{S}^2$.

On $\mathbb{T}^2$, we compare small Moser Flows, DDPM and EGFs, demonstrating that EGFs can generate well-behaved distributions even when Moser Flows break down. An EGF is built with the minimal four affine transformations described in section 3.1.2, implemented as MLPs with 3 hidden layers of width 32. These transformations parameterize $f_\to^*$ and $\pi_\to^*$. The Moser Flow is trained using the implementation provided in the authors' GitHub repository (Rozen, 2022) with the only modification being the model size set to 32x3. Our EGF is trained using the KL-weakFM loss together with

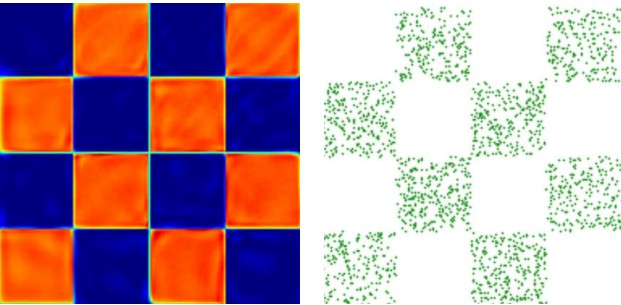

a EGF 32x5, density (left) and samples (right).

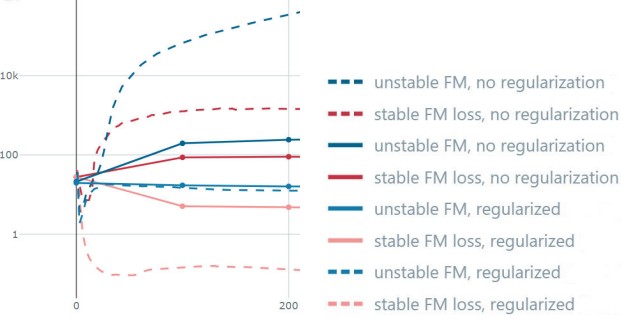

b EGF 32x5, stability comparison between FM losses. Unregularized unstable loss (blue) flow size (dashed lines) and sampling time $\tau$ (solid lines) blow up while stable loss (red) converges. Regularization helps stabilization for both losses.

*Figure 1.* Checkerboard RL task.

*Table 1.* Negative log-likelihood scores of the volcano dataset.

|  | Volcano | Earthquake | Flood |
|---|---|---|---|
| Mixture vMF | -0.31 | 0.59 | 1.09 |
| Stereographic | -0.64 | 0.43 | 0.99 |
| Riemannian | -0.97 | 0.19 | 0.90 |
| Moser Flow | -2.02 | -0.09 | 0.62 |
| EGFN | **-2.31** | **-0.12** | **0.56** |

a regularization $\mathcal{R}$ as in the RL setting. Figure 10 shows how Moser Flow fails to train with such a small model while minimal EGF reproduces the target distribution with high fidelity.

On $\mathbb{S}^2$, we benchmark EGFs on the earth science volcano dataset (NGDC/WDS, 2025). A sample distribution is given for the dataset in Figure 3 and negative log-likelihoods are given in Table 1. We only use six rotations: a rotation of angle $\pi/4$ around each of the three axes plus their respective inverse. The two core MLPs of EGF are of size 256x5, compared to the 512x6 used by Rozen et al. (2021). The learning rate is 1e-3 with an exponential decay to 1e-5 at 3000 epochs of 25 steps. We outperform Moser Flow and all related baselines. Notably, the EGF achieved its reported performance with a training time 10 times shorter.

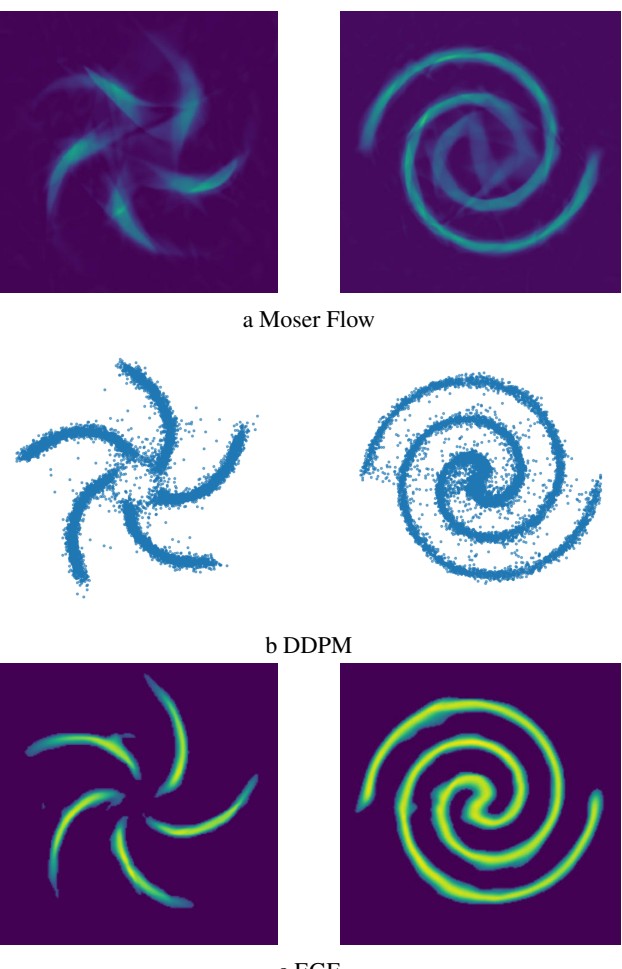

a Moser Flow

b DDPM

c EGF

*Figure 2.* Comparison of imitation learning on standard toy distributions using tiny 32x3 MLP models. Background filter is applied to EGF, a similar filter on Moser flow would yield worse results. Without a filter, EGF samples have outliers similar to DDPM. Since DDPM does not give access to density, only samples are provided. For fairness, DDPM is sampled using 100 denoising steps.

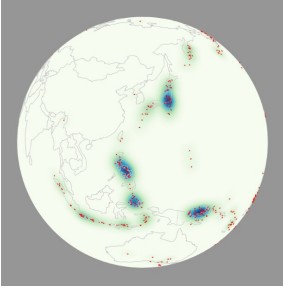
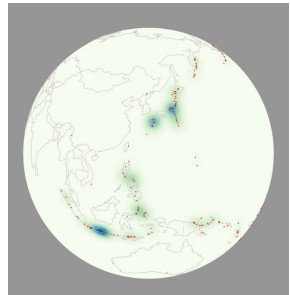

*Figure 3.* EGF generated points with reward field $\widehat{f}_{\text{term}}$ (left) and whole dataset point with KDE field (right) for the volcano task.

## 5. Related Work

EGFs draw inspiration from NFs for their transformation blocks, as well as from denoising diffusion models (DDMs) (Ho et al., 2020) for incorporating stochasticity. While similar approaches have been explored in previous works (Wu et al., 2020; Zhang & Chen, 2021), our method stands out by leveraging general generative flows theory to construct a more efficient framework. Compared to previous works, EGFs are more compact and demonstrate faster convergence as illustrated in our experiments. Furthermore, DDMs and their variants (Song et al., 2020a) rely on a fixed discretization of inference denoising trajectories, whereas EGFs enable trainable sampling times, offering greater flexibility and adaptability.

A particularly notable comparison arises when examining the relationship between our EGFs and NFs introduced in Rezende et al. (2020), as both utilize affine toroidal families and projective spherical families. However, while our EGF framework provides universality guarantees, the NF setting lacks such guarantees as it solely relies on affine and projective transforms. Indeed, both projective and affine transformations form groups (compositions of affine transforms are affine), thus limiting the expressivity. This distinction highlights a fundamental advantage of EGFs, further reinforcing their theoretical robustness and practical effectiveness.

Broadly speaking, previous methods, such as Riemannian Continuous Normalizing Flows (Mathieu & Nickel, 2020) and Flow Matching on Manifolds (Chen & Lipman, 2024) rely on either handcrafted non-linear transformations or sophisticated NeuralODEs (Chen et al., 2018). Our EGF does not incorporate NeuralODEs, as we do not face expressivity limitations that justify their usage. However, these techniques remain fully compatible with EGFs and could be integrated into future extensions of our framework to further enhance its flexibility and adaptability if needed.

Since EGFs' elementary transformation are derived from the transformation blocks of NFs, EGFs can be viewed as a stochastic sampler for NF architectures. In particular, any EGFs sampled trajectory can be defined as $(s_t)_{t=1}^\tau$, where the trajectory $(s_t)_{t=1}^\tau$ is obtained by composing NF transformations as follows: $s_1 \sim \mathcal{F}_{\text{init}}$, $s_2 = \Phi_{k_1}(s_1)$, $\ldots$, $s_\tau = \Phi_{k_{\tau-1}} \circ \cdots \circ \Phi_{k_1}(s_1)$, with the transition probability $\mathbb{P}(k_{t+1} \mid k_t) = \alpha_\rightarrow^i(s_t)$. Each composition $\Phi_{k_{\tau-1}} \circ \cdots \circ \Phi_{k_1}$ is a diffeomorphism, and the terminal distribution is given by $F_{\text{term}} = \mathbb{E}_K(F_{\text{init}}K)$, where $K = \Phi_{k_{\tau-1}} \circ \cdots \circ \Phi_{k_1}$ represents a random NF. In other words, an EGF with $p$ diffeomorphisms can be represented as a pair $(\iota, \pi)$, where $\iota : \mathcal{T}_p \to \text{Diffeo}(\mathcal{S})$ is a trainable embedding of the $p$-ary tree into the space of diffeomorphisms on the state space, and $\pi$ is a random walk policy on $\mathcal{T}_p$ with a random stopping time. However, the policy $\pi$ is intractable as it is defined by

$\pi(\Phi_k|\Phi_{k_t} \circ \cdots \Phi_{k_1}) = \frac{1}{F_\rightarrow(\mathcal{S})} \int_{s \in \mathcal{S}} \alpha_\rightarrow^k(s) dF_\rightarrow(s)$, where $F_\rightarrow := F_\rightarrow^* + F_{\text{term}}$.

Building on this formulation, we can draw comparisons between EGFs and three related research directions: Continuously Indexed Normalizing Flows (CINF) (Caterini et al., 2021), Neural Architecture Search (NAS) (Elsken et al., 2019), and Wasserstein Gradient Descent (WGD) (Chizat & Bach, 2018). First, CINFs attempt to overcome the limitations of NFs by using a fixed architecture, with conditioning sampled from a latent distribution. This approach generates the target distribution as an expectation, $F_{\text{term}} = \mathbb{E}_K(F_{\text{init}}K)$, where the random NF $K$ is drawn from a continuous distribution of NFs. In contrast, our method samples $K$ from a tree structure. Second, NAS aims to find the optimal neural network architecture for a given task. In comparison, our EGF constructs a distribution of suitable NFs, where the sampling time can be interpreted as a learned depth of the resulting architecture. Third, by training over a distribution of architectures and considering the expectation of the output (with $F_{\text{init}}$ as the input and $F_{\text{term}}$ as the output), our approach parallels the setup of Wasserstein Gradient Descent (WGD). Extending the theorems of Chizat & Bach (2018) to EGFs would be a valuable addition to our theoretical framework.

Lastly, from the proof of the universality Theorem 3.4, we observe that $f_\rightarrow^*$ can be obtained as a fixed point of a flow operator dependent on $\pi_\rightarrow^*$, $F_{\text{init}}$, and $F_{\text{term}}$. This operator is a contraction for the affine toroidal family. Although we train $f_\rightarrow^*$ via gradient descent, this approach bears similarities to Deep Equilibrium Models (DEQ) (Bai et al., 2019; 2020), where the fixed point is computed by a contraction neural network. The key difference is that we explicitly approximate the fixed point with a neural network, while DEQ models use a black-box solver and the implicit function theorem for gradients. Future work could develop a DEQ-EGF, where $f_\rightarrow^*$ is implicitly obtained as a fixed point instead of via a feedforward network.

## 6. Limitations and Future Work

First and foremost, our experiments are limited to low dimensions. Although ergodicity is easily achievable with a number of generators independent from the dimension (for instance on tori, one randomly initialized non-trainable translation is sufficient to ensure ergodicity), the technical assumption of $L^2$-mixing summability needed for universality is more subtle. EGFs suffer from limited proven applicability, further theory is needed to easily enforce $L^2$-mixing summability in higher dimension together with related experiments. More precisely, a theoretical bound on the minimal number of transformations necessary to achieve universality would be useful for hyperparameter tuning. We give such a bound only for affine toroidal and isometry

spherical families where two transformations are sufficient to achieve universality.

Second, the $L^2$-mixing summability property is essential to ensure universality. It is likely that training would be improved if some control over this property was to be achieved, say by adding a regularization. There is extensive mathematical literature on the so-called spectral gap (Kontoyiannis & Meyn, 2012; Guivarc'h & Le Page, 2016; Marrakchi, 2018; Bekka & Francini, 2020), which is a sufficient condition for $L^2$-mixing summability, and absolutely continuous invariant measures (Góra & Boyarsky, 2003; Bahsoun, 2004; Galatolo, 2015) including stability results (Froyland et al., 2014). A systematic review of this literature would certainly yield such regularization.

Lastly, our focus was on the theoretical advancements of generative flow theory through EGF. As a result, we did not exploit EGF's high modularity, which enables sophisticated transformations, replay buffers, or neural architectures. Additionally, hyperparameter tuning was minimal, leaving room for future work to perform a systematic analysis.

## 7. Conclusion

We propose a new family of generative flows that leverage ergodicity to provide universality guarantees while utilizing simple diffeomorphisms and neural networks. Our results demonstrate that EGFs maintain their expressivity even at parameter counts where Moser Flow (Rozen et al., 2021) fails. This simplicity enables us to outperform our baselines on the NASA volcano dataset with a model 30 times smaller.

## Impact Statement

This paper presents work whose goal is to advance the field of Machine Learning. There are many potential societal consequences of our work, none which we feel must be specifically highlighted here.

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

# A. Fundamentals of EGFs

,

As stated in section 3, on a Differential manifold $\mathcal{S}$ endowed with a background measure $\lambda$ absolutely continuous with respect to Lebesgue measure, we define an EGF as a Generative Flow $(\pi_\rightarrow^*, F_\rightarrow^*)$ with $F_\rightarrow^* = f_\rightarrow^* \lambda$ and $\pi_\rightarrow^*(s \rightarrow \cdot) = \sum_{i=1}^p \alpha_\rightarrow^i(s)\delta_{\Phi_i(s)}$ where $\Phi_i$ are diffeomorphisms of $\mathcal{S}$. With these definitions, the backward policy automatically has a closed form formula.

**Theorem A.1.** *Let* $(\pi_\rightarrow^*, F_\rightarrow^*)$ *as above, the induced Generative Flow on* $\mathcal{S}$ *has a backward star policy given by*

$$\pi_\leftarrow^*(s) = \sum_{i=1}^p \alpha_\leftarrow^i(s)\delta_{\Phi_i^{-1}(s)} \tag{21}$$

$$\alpha_\leftarrow^i(s) = \frac{(\alpha_\rightarrow^i f_\rightarrow^*) \circ \Phi_i^{-1}(s)|J_s\Phi_i^{-1}|}{f_\leftarrow^*(s)} \tag{22}$$

$$f_\leftarrow^* = \sum_j (\alpha_\rightarrow^j f_\rightarrow^*) \circ \Phi_j^{-1}(s)|J_s\Phi_j^{-1}|. \tag{23}$$

*Proof.* From (Brunswic et al., 2024) section 3.2 and Proposition 1 in their appendix, we have for any measurable $X \subset \mathcal{S}$:

$$F_\leftarrow^*(Y) = F_\rightarrow^* \otimes \pi_\rightarrow^*(\mathcal{S} \rightarrow Y) \tag{24}$$

$$\pi_\leftarrow^*(s \rightarrow X) = \frac{dF_\leftarrow^* \otimes \pi_\leftarrow^*(\cdot \rightarrow X)}{dF_\leftarrow^*}(s) \tag{25}$$

$$F_\leftarrow^* \otimes \pi_\leftarrow^*(Y \rightarrow X) = F_\rightarrow^* \otimes \pi_\rightarrow^*(X \rightarrow Y). \tag{26}$$

Therefore, for all measurables $X, Y \subset \mathcal{S}$:

$$F_\leftarrow^* \otimes \pi_\leftarrow^*(Y \rightarrow X) = F_\rightarrow^* \otimes \pi_\rightarrow^*(X \rightarrow Y) \tag{27}$$

$$= \int_{s \in \mathcal{S}} \mathbf{1}_X(s)\pi_\rightarrow^*(s \rightarrow Y)dF_\rightarrow^*(s) \tag{28}$$

$$= \int_{s \in \mathcal{S}} \mathbf{1}_X(s)\pi_\rightarrow^*(s \rightarrow Y)f_\rightarrow^*(s)d\lambda(s) \tag{29}$$

$$= \int_{s \in \mathcal{S}} \sum_{i=1}^p \mathbf{1}_X(s)\alpha_\rightarrow^i(s)\delta_{\Phi_i(s)}(Y)f_\rightarrow^*(s)d\lambda(s) \tag{30}$$

$$= \sum_{i=1}^p \int_{s \in \mathcal{S}} \mathbf{1}_X(s)\alpha_\rightarrow^i(s)\mathbf{1}_Y(\Phi_i(s))f_\rightarrow^*(s)d\lambda(s) \tag{31}$$

$$= \sum_{i=1}^p \int_{u \in \mathcal{S}} \mathbf{1}_X(\Phi_i^{-1}(u))\alpha_\rightarrow^i(\Phi_i^{-1}(u))\mathbf{1}_Y(u)f_\rightarrow^*(\Phi_i^{-1}(u))|J_u\Phi_i^{-1}|d\lambda(u) \tag{32}$$

$$= \sum_{i=1}^p \int_{u \in Y} \mathbf{1}_X(\Phi_i^{-1}(u))\alpha_\rightarrow^i(\Phi_i^{-1}(u))f_\rightarrow^*(\Phi_i^{-1}(u))|J_u\Phi_i^{-1}|d\lambda(u) \tag{33}$$

$$= \int_{u \in Y} \sum_{i=1}^p \delta_{\Phi_i^{-1}(u)}(X)\alpha_\rightarrow^i(\Phi_i^{-1}(u))f_\rightarrow^*(\Phi_i^{-1}(u))|J_u\Phi_i^{-1}|d\lambda(u). \tag{34}$$

$\square$

Since this formula is true for any $Y \subset \mathcal{S}$, we deduce that $F_{\leftarrow}^* \otimes \pi_{\leftarrow}^*(\cdot \to X) \ll \lambda$ and that

$$\frac{dF_{\leftarrow}^* \otimes \pi_{\leftarrow}^*(\cdot \to X)}{d\lambda}(x) = \sum_{i=1}^{p} \delta_{\Phi_i^{-1}(x)}(X) \alpha_{\to}^i(\Phi_i^{-1}(x)) f_{\to}^*(\Phi_i^{-1}(x)) |J_x \Phi_i^{-1}| \tag{35}$$

$$= \left( \sum_{i=1}^{p} \alpha_{\to}^i(\Phi_i^{-1}(x)) f_{\to}^*(\Phi_i^{-1}(x)) |J_x \Phi_i^{-1}| \delta_{\Phi_i^{-1}(x)} \right)(X) \tag{36}$$

$$f_{\leftarrow}^*(x) = \frac{dF_{\leftarrow}^* \otimes \pi_{\leftarrow}^*(\cdot \to \mathcal{S})}{d\lambda}(x) \tag{37}$$

$$= \sum_{i=1}^{p} \delta_{\Phi_i^{-1}(x)}(\mathcal{S}) \alpha_{\to}^i(\Phi_i^{-1}(x)) f_{\to}^*(\Phi_i^{-1}(x)) |J_x \Phi_i^{-1}| \tag{38}$$

$$= \sum_{i=1}^{p} \alpha_{\to}^i(\Phi_i^{-1}(x)) f_{\to}^*(\Phi_i^{-1}(x)) |J_x \Phi_i^{-1}| \tag{39}$$

The result follows.

# B. Universality Theorems

## B.1. General Universality Theorem

We shall both write the theorems in a more formal way and provide rigorous proof of each statements.

The following theorem implies Theorem 3.4.

**Theorem B.1.** *Let $(\mathcal{S}, \lambda)$ be a measured Polish space and let $\pi_{\to}^*$ be a Markov kernel acting on $\mathcal{S}$. Assume that $\pi_{\to}^*$ is ergodic on $(\mathcal{S}, \lambda)$ with summable $L^2$-mixing coefficients, then for any probability distributions $F_{\text{init}}, F_{\text{term}} \ll \lambda$ with density in $L^2(\lambda)$ and any $\varepsilon > 0$, there exists $f_{\to}^* \in L^2(\lambda)$ such that the generative flow $(\pi_{\to}^*, f_{\to}^*)$ from $F_{\text{init}}$ to $F_{\text{term}}$ is such that $\delta F_{\text{init}}(\mathcal{S}) + \delta F_{\text{term}}(\mathcal{S}) < \varepsilon$.*

*Proof.* Let $H : \nu \mapsto \nu \pi^* + F_{\text{init}} - F_{\text{term}}$, let $\nu_0 = F_{\text{init}} - F_{\text{term}}$ and let $\nu_{t+1} = H(\nu_t)$. By assumption, $\nu_0$ density is in $L^2(\lambda)$, then the density of $\nu_t$ with respect to $\lambda$ is $L^2$ for all $t \in \mathbb{N}$.

We have

$$\nu_T = H^T(\nu) \tag{40}$$

$$= \sum_{t=0}^{T} (F_{\text{init}} - F_{\text{term}})(\pi_{\to}^*)^t \tag{41}$$

$$= \sum_{t=0}^{T} \epsilon_t \tag{42}$$

with $\epsilon_t = (F_{\text{init}} - F_{\text{term}})(\pi_{\to}^*)^t$. For all $t \in \mathbb{N}, \epsilon_t(\mathcal{S}) = 0$, hence by assumption $\sum_{t \geq 0} \int_{s \in \mathcal{S}} \left( \frac{d\epsilon_t}{d\lambda}(s) \right)^2 d\lambda(s) < +\infty$. Therefore, $\nu_T$ converges as $T \to +\infty$ to some $\nu_\infty$ and $|\nu_\infty|_2 \leq \sum_{t \geq 0} \int_{s \in \mathcal{S}} \left( \frac{d\epsilon_t}{d\lambda}(s) \right)^2 d\lambda(s) < +\infty$, hence $\nu_\infty \ll \lambda$ and $\frac{d\nu_\infty}{d\lambda} \in L^2(\lambda)$. Furthermore, $\nu_\infty$ is a fix point of $H$ so $\nu_\infty = \nu_\infty \pi^* - R + F_{\text{init}}$.

Now, $\nu_\infty$ does not necessarily provide a suitable $f_{\to}^*$ because it may happen that the negative part $\nu_\infty^- \neq 0$. Since for any $\eta > 0$ we have $H(\nu_\infty + \eta\lambda) = H(\nu_\infty) + \eta\lambda\pi_{\to}^* = \nu_\infty + \eta\lambda$, we may add $\eta\lambda$ to $\nu_\infty$ to get another fix point $\nu_\infty^\eta := \nu_\infty + \eta\lambda$ of $H$. Define $f_{\to}^* = \frac{d(\nu_\infty^\eta)^+}{d\lambda}$, so that:

$$F_{\text{init}} + (f_{\to}^*\lambda)\pi_{\to}^* - F_{\text{term}} - f_{\to}^*\lambda = F_{\text{init}} + (f_{\to}^*\lambda - \nu_\infty^\eta + \nu_\infty^\eta)\pi_{\to}^* - F_{\text{term}} - (f_{\to}^*\lambda - \nu_\infty^\eta + \nu_\infty^\eta) \tag{43}$$

$$= (f_{\to}^*\lambda - \nu_\infty^\eta)\pi_{\to}^* - (f_{\to}^*\lambda - \nu_\infty^\eta) \tag{44}$$

Therefore, $\delta F_{\text{init}} := [(f_{\to}^*\lambda - \nu_\infty^\eta)\pi_{\to}^* - (f_{\to}^*\lambda - \nu_\infty^\eta)]^-$ and $\delta F_{\text{term}} := [(f_{\to}^*\lambda - \nu_\infty^\eta)\pi_{\to}^* - (f_{\to}^*\lambda - \nu_\infty^\eta)]^+$. So that

$$\delta F_{\text{init}}(\mathcal{S}) + \delta F_{\text{term}}(\mathcal{S}) = |(\nu_\infty^\eta)^-(\pi_{\to}^* - 1)|(\mathcal{S}) \leq 2(\nu_\infty^\eta)^-(\mathcal{S}). \tag{45}$$

Since $\nu_\infty \ll \lambda$ in particular $\lim_{\eta \to +\infty} (\nu_\infty^\eta)^-(\mathcal{S}) = 0$, the result follows by choosing $\eta$ big enough. $\qquad \square$

## B.2. Universality on Tori

The universality Theorem on tori is obtained as a consequence of the spectral gap for Affine Toroidal families. We use the following result.

**Theorem B.2** (Theorem 5 of (Bekka & Guivarc'h, 2015) ). *Let $H$ be a countable subgroup of $\mathrm{Aff}(\mathbb{T}^d)$. The following properties are equivalent:*

  (i) *The action of $H$ on $\mathbb{T}$ does not have a spectral gap.*

  (ii) *There exists a non-trivial $H$-invariant factor torus $\overline{\mathbb{T}}$ such that the projection of $H$ on $\mathrm{Aut}(\overline{\mathbb{T}})$ is amenable.*

If condition $(ii)$ of Theorem B.2 is false for the group generated by the transformations $(\Phi_i)_{i=1}^p$, the we deduce any $\pi_\rightarrow^*$ induced by equation (5) has spectral gap if $\forall i, \alpha_\rightarrow^i = \frac{1}{p}$. Since each $\Phi_i$ keep the Lebesgue measure $\lambda$ invariant, by universality master Theorem B.1, the family is universal. Our technical condition of Theorem 3.5 is then "The set of transforms $(\Phi_i)_{i=1}^p$ is stable by inverse and the group generated by $(\Phi_i)_{i=1}^p$ violates condition $(ii)$." In particular, if the linear part of $H$ is the whole $\mathrm{SL}(d,\mathbb{Z})$, projections such as the one in condition $(ii)$ are never amenable since $\mathrm{SL}(d,\mathbb{Z})$ contains a free group with two generators.

## B.3. Universality on Spheres

The universality Theorem on spheres is obtained as a consequence of the spectral gap for isometry spherical families. Before stating the technical mathematical result we use, we recall that a real number $x \in \mathbb{R}$ is *algebraic* if there exists a non-zero polynomial $P \in \mathbb{Z}[X]$ with integral coefficients such that $P(x) = 0$. The set of algebraic real number is denoted $\overline{\mathbb{Q}}$. We use the following result.

**Theorem B.3** (Reformulation of Theorem 1 of (Bourgain & Gamburd, 2012) ). *Assume that $(\Phi_1, \cdots, \Phi_p) \in \mathrm{SO}(d) \cap \mathrm{Mat}_{d \times d}(\overline{\mathbb{Q}})$, that the group generated by $\Phi_1, \cdots, \Phi_p$ is dense in $\mathrm{SO}(d,\mathbb{R})$ and that $\forall i \in \{1, \cdots, p\}, \exists j \in \{1, \cdots, p\}, \Phi_i^{-1} = \Phi_j$. Assume that, $\forall i, \alpha_\rightarrow^i = \frac{1}{p}$ then the associated Markov kernel $\pi_\rightarrow^*$ given by equation (5) has a spectral gap.*

With the technical condition "The set of transforms $(\Phi_i)_{i=1}^p$ is stable by inverse, the group generated by $(\Phi_i)_{i=1}^p$ is dense in $\mathrm{SO}(d,\mathbb{R})$ and each of the $\Phi_i$ has algebraic coefficients", Theorem 3.6 then follows from master Theorem B.1.

# C. Quantitative Sampling Theorem

**Theorem C.1.** *Let $(\pi_\rightarrow^*, f_\rightarrow^*)$ be a generative flow and let $F_{\mathrm{init}}$ and $F_{\mathrm{term}}$ initial and terminal distributions. Assume that $F_{\mathrm{init}} \neq 0$, and consider the sample $s_\tau$ of the generative flow Markov chain from $F_{\mathrm{init}}$ to $\widehat{F}_{\mathrm{term}}$ then:*

$$\mathrm{TV}\left(s_\tau \left\| \frac{1}{\widehat{F}_{\mathrm{term}}(\mathcal{S})} \widehat{F}_{\mathrm{term}} \right.\right) \leq \frac{\delta F_{\mathrm{init}}(\mathcal{S})}{\widehat{F}_{\mathrm{term}}(\mathcal{S})} \tag{46}$$

*with* TV *the total variation.*

*Proof.* For any generative flow $\mathbb{F} := (\pi_\rightarrow^*, F_\rightarrow^*)$, define $\pi_\rightarrow^{\tau, \mathbb{F}}$ the kernel $x \mapsto (s_\tau | s_1 = x)$ for the stopping condition $\widehat{F}_{\mathrm{term}}$. The sampling distribution $s_\tau$ of $\mathbb{F}$ is then $\frac{1}{F_{\mathrm{init}}(\mathcal{S})} F_{\mathrm{init}} \pi_\rightarrow^{\tau, \mathbb{F}}$. Furthermore, we have $\delta F_{\mathrm{term}}(\mathcal{S}) - \delta F_{\mathrm{init}}(\mathcal{S}) = F_{\mathrm{init}}(\mathcal{S}) + F_\rightarrow^* \pi^*(\mathcal{S}) - F_\rightarrow^*(\mathcal{S}) - F_{\mathrm{term}}(\mathcal{S}) = F_{\mathrm{init}}(\mathcal{S}) - F_{\mathrm{term}}(\mathcal{S})$. In particular, $\widehat{F}_{\mathrm{term}}(\mathcal{S}) \geq F_{\mathrm{init}}(\mathcal{S}) > 0$ so $\widehat{F}_{\mathrm{term}} \neq 0$.

We notice that the generative flow $\mathbb{F}' := (\pi_\rightarrow^*, F_\rightarrow^*)$ satisfies the flow-matching constraint from $\widehat{F}_{\mathrm{init}}$ to $\widehat{F}_{\mathrm{term}}$ and that $\pi^{\tau, \mathbb{F}'} = \pi^{\tau, \mathbb{F}}$. Applying the sampling Theorem to $\mathbb{F}'$, we obtain

$$\widehat{F}_{\mathrm{init}} \pi_\rightarrow^{\tau, \mathbb{F}'} = \widehat{F}_{\mathrm{term}} \tag{47}$$

$$(F_{\mathrm{init}} + \delta F_{\mathrm{init}}) \pi_\rightarrow^{\tau, \mathbb{F}'} = \widehat{F}_{\mathrm{term}} \tag{48}$$

$$F_{\mathrm{init}} \pi_\rightarrow^{\tau, \mathbb{F}'} + \delta F_{\mathrm{init}} \pi_\rightarrow^{\tau, \mathbb{F}'} = \widehat{F}_{\mathrm{term}} \tag{49}$$

$$\alpha \times \underbrace{\left( \frac{1}{F_{\mathrm{init}}(\mathcal{S})} F_{\mathrm{init}} \pi_\rightarrow^{\tau, \mathbb{F}} \right)}_{s_\tau} + (1 - \alpha) \times \frac{1}{\delta F_{\mathrm{init}}(\mathcal{S})} \delta F_{\mathrm{init}} \pi_\rightarrow^{\tau, \mathbb{F}'} = \frac{1}{\widehat{F}_{\mathrm{term}}(\mathcal{S})} \widehat{F}_{\mathrm{term}} \tag{50}$$

with $\alpha = \frac{F_{\text{init}}(\mathcal{S})}{\widehat{F}_{\text{term}}(\mathcal{S})} = \frac{\widehat{F}_{\text{term}}(\mathcal{S}) - \delta F_{\text{init}}(\mathcal{S})}{\widehat{F}_{\text{term}}(\mathcal{S})} = 1 - \frac{\delta F_{\text{init}}(\mathcal{S})}{\widehat{F}_{\text{term}}(\mathcal{S})}$. The result follows. $\qquad\square$

## D. Implementation considerations

### D.1. Density filtering

Since EGFs give us access to a tractable density $\widehat{f}_{\text{term}}$, we can estimate its mean $m$ and standard deviation $\sigma$ on the state space to filter out any unwanted outliers when sampling. By choosing a lower bound saturation value $\widehat{f}_{sat} = m - k\sigma$ where $k$ defines how strong the filter should be, we can then define a stricter sampling density $\tilde{f}_{\text{term}}$ as:

$$\tilde{f}_{term} = \begin{cases} \widehat{f}_{\text{term}} & \text{if } \widehat{f}_{\text{term}} \geq \widehat{\tilde{f}}_{\text{term}} \\ 0 & \text{otherwise} \end{cases} \tag{51}$$

Since the new sampling density $\tilde{f}_{\text{term}} = 0$ for all points of low true density, the next state of the Markov Chain is guaranteed not be the sink state. This ensures the EGF only generates samples associated with high density. The factor $k$ can be chosen manually to obtain a more strict or lenient filter, or it can be automatically optimized by recalculating the negative log-likelihood on the training dataset to systematically evaluate all $k$ values.

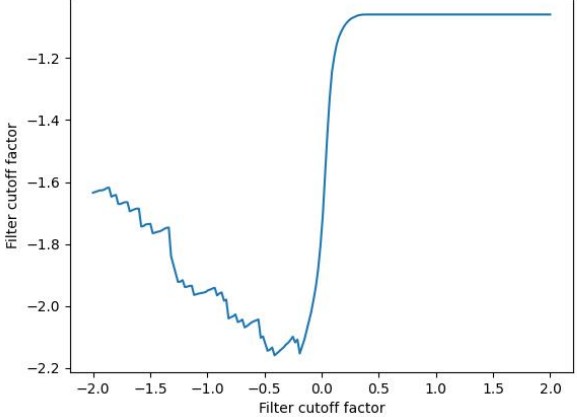

*Figure 4.* NLL estimation for a range of values of $k$ for a given model on the volcano dataset.

Note that this filtering is applied on the density rather than on the samples, meaning that no samples are filtered out. Therefore, this simply has the effect of preventing the trajectories from stopping at low density points. Figure 5 demonstrates that using the optimal filter value concentrates the samples in the zones of high true density, bypassing the uniform background noise that would otherwise be found prior to filtering.

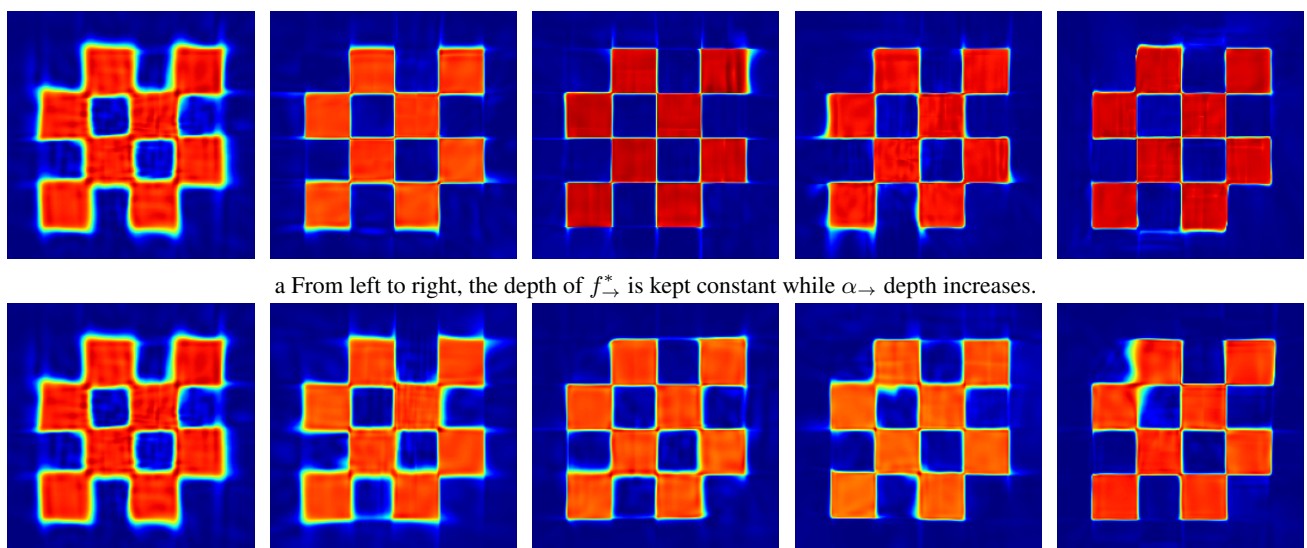

a From left to right, the depth of $f_\rightarrow^*$ is kept constant while $\alpha_\rightarrow$ depth increases.

b From left to right, the depth of $\alpha_\rightarrow$ is kept constant while $f_\rightarrow^*$ depth increases.

*Figure 6.* Variation of $f_\rightarrow^*$ and $\alpha_\rightarrow$ depths from 2x32 to 6x32. The theoretical prediction is that $\alpha_\rightarrow$ may be kept constant while $f_\rightarrow^*$ is trained. Keeping $\alpha_\rightarrow$ constant indeed yields improvements, however the improvement obtained by increasing the depth of $\alpha_\rightarrow$ instead of $f_\rightarrow^*$ is comparatively better.

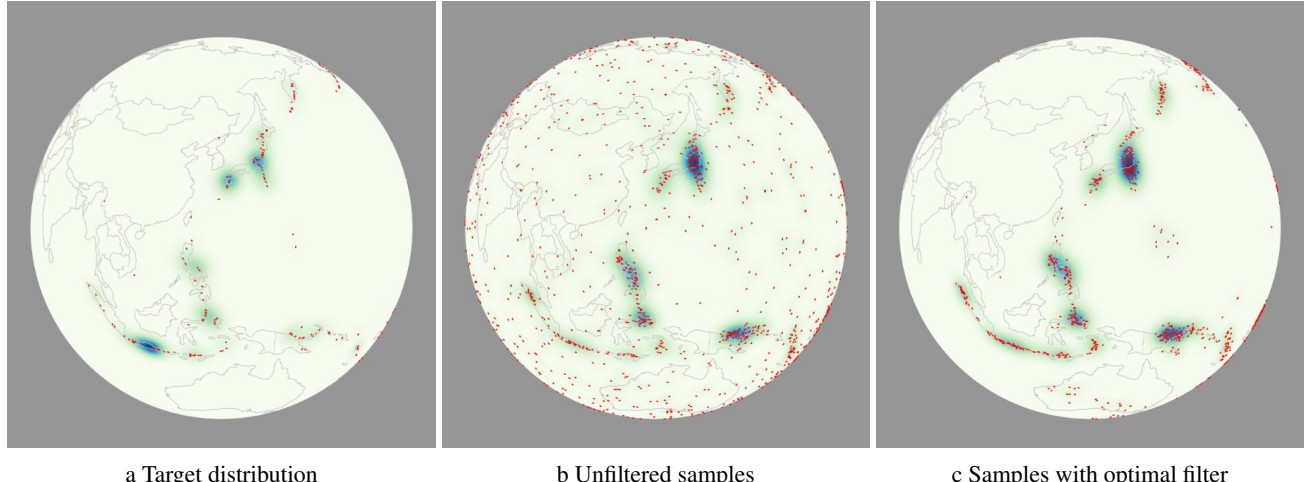

a Target distribution        b Unfiltered samples        c Samples with optimal filter

*Figure 5.* Comparison of the EGF's samples using the same model with and without a density filter.

# E. Ablation study on the size of EGF

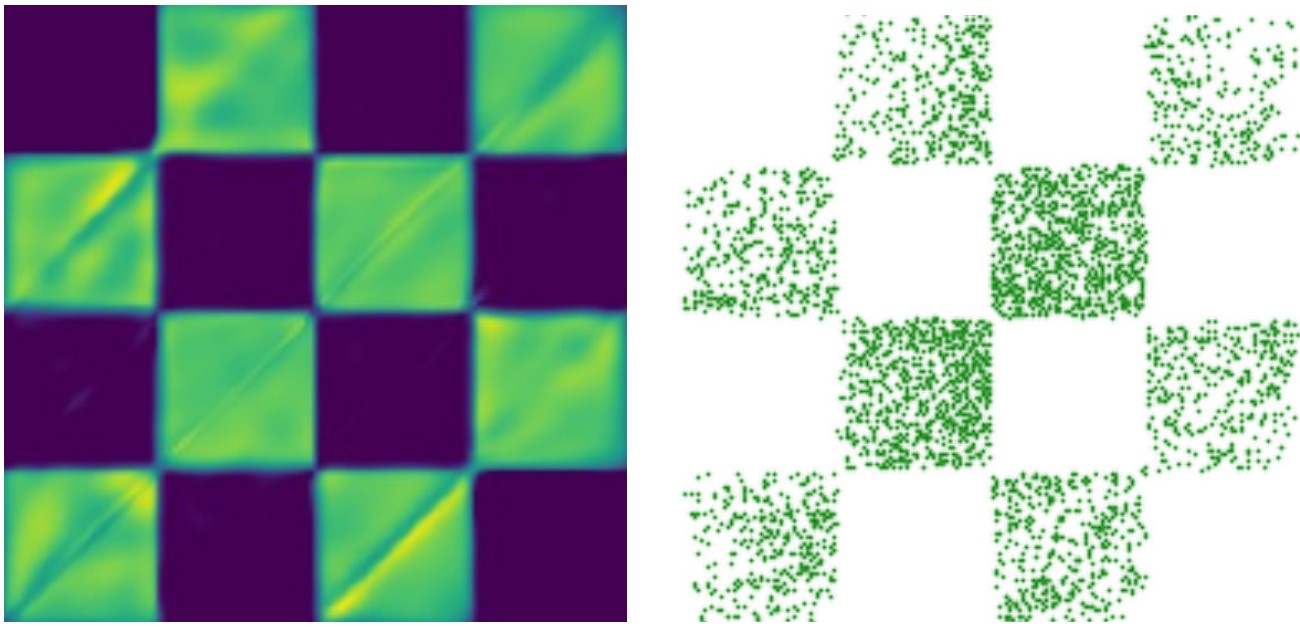

*Figure 7.* Theoretically minimal EGF with four affine moves on the torus $\mathbb{T}^2$ ie two generators of $\mathrm{SL}(2, \mathbb{Z})$ together with their inverse. The MLPs parameterizing $f_{\rightarrow}^*$ and $\alpha_{\rightarrow}$ have size 4x128. We see that the generated checkerboard has high quality, sharp boundaries but some under densities

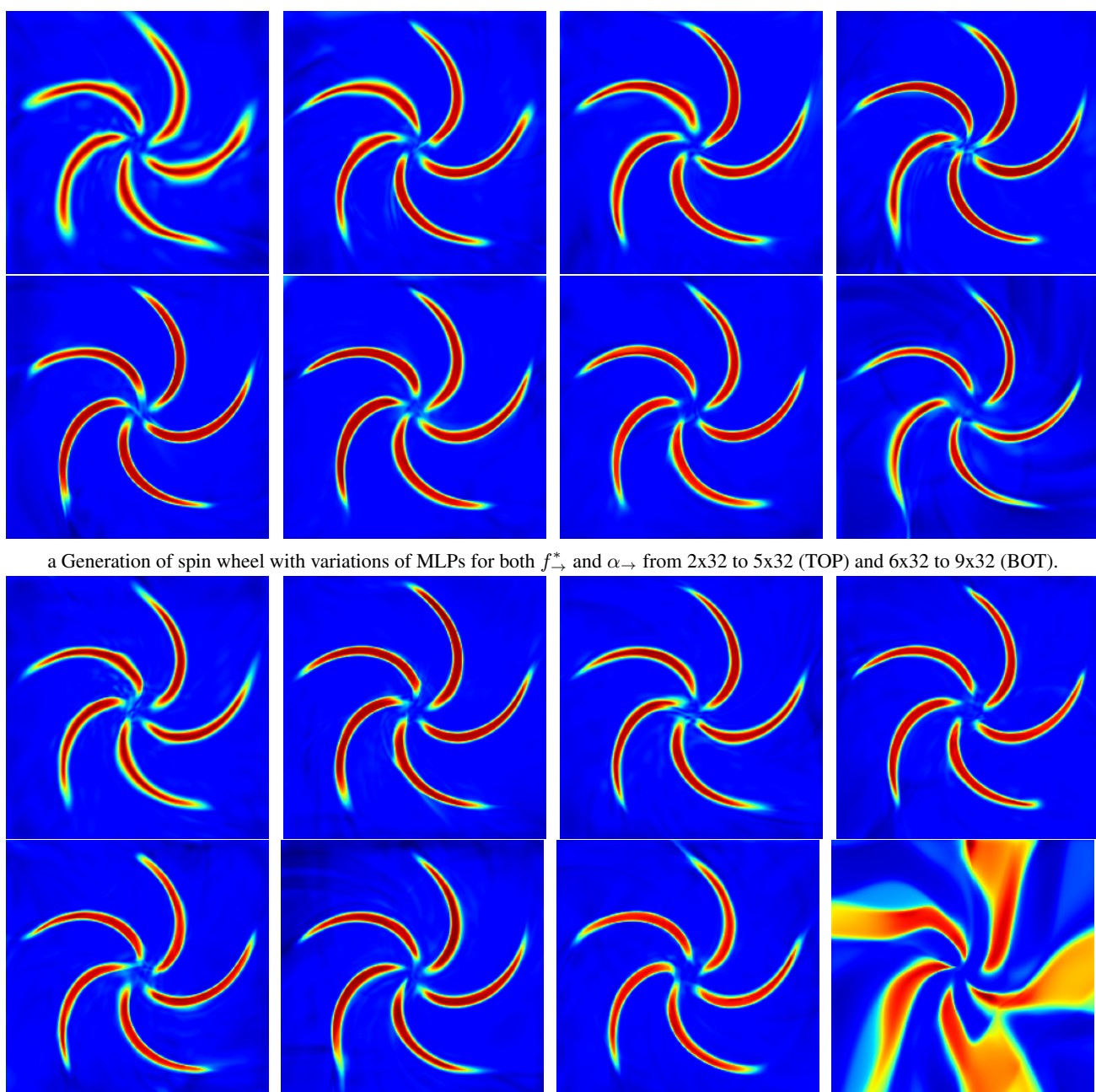

a Generation of spin wheel with variations of MLPs for both $f_{\rightarrow}^{*}$ and $\alpha_{\rightarrow}$ from 2x32 to 5x32 (TOP) and 6x32 to 9x32 (BOT).

b Generation of spin wheel with variations of MLPs for both $f_{\rightarrow}^{*}$ and $\alpha_{\rightarrow}$ from 2x64 to 5x64 (TOP) and 6x64 to 9x64 (BOT).

*Figure 8.* Variation of $f_{\rightarrow}^{*}$ and $\alpha_{\rightarrow}$ depths and width from 2x32 to 9x64. Increasing depth and width increases quality until depth 4 then the training seemingly becomes lest stable as shown by the 9x64 experiment. Since we keep the learning rate constant at 1e-3, it is likely that the MLPs training becomes unstable. The state space is $\mathcal{S} = \mathbb{R}^2$, to ensure $F_{\rightarrow}(\mathcal{S}) < +\infty$, we constrain $f_{\rightarrow}^{*}$ to be zero outside $[-4, 4]^2$. The EGF has 8 transformations which are translations, while the MLPs for $\alpha_{\rightarrow}$ and $f_{\rightarrow}^{*}$ vary from 2x32 (2 hidden layer of width 32) to 9x64. Learning rate is kept at 0.001.

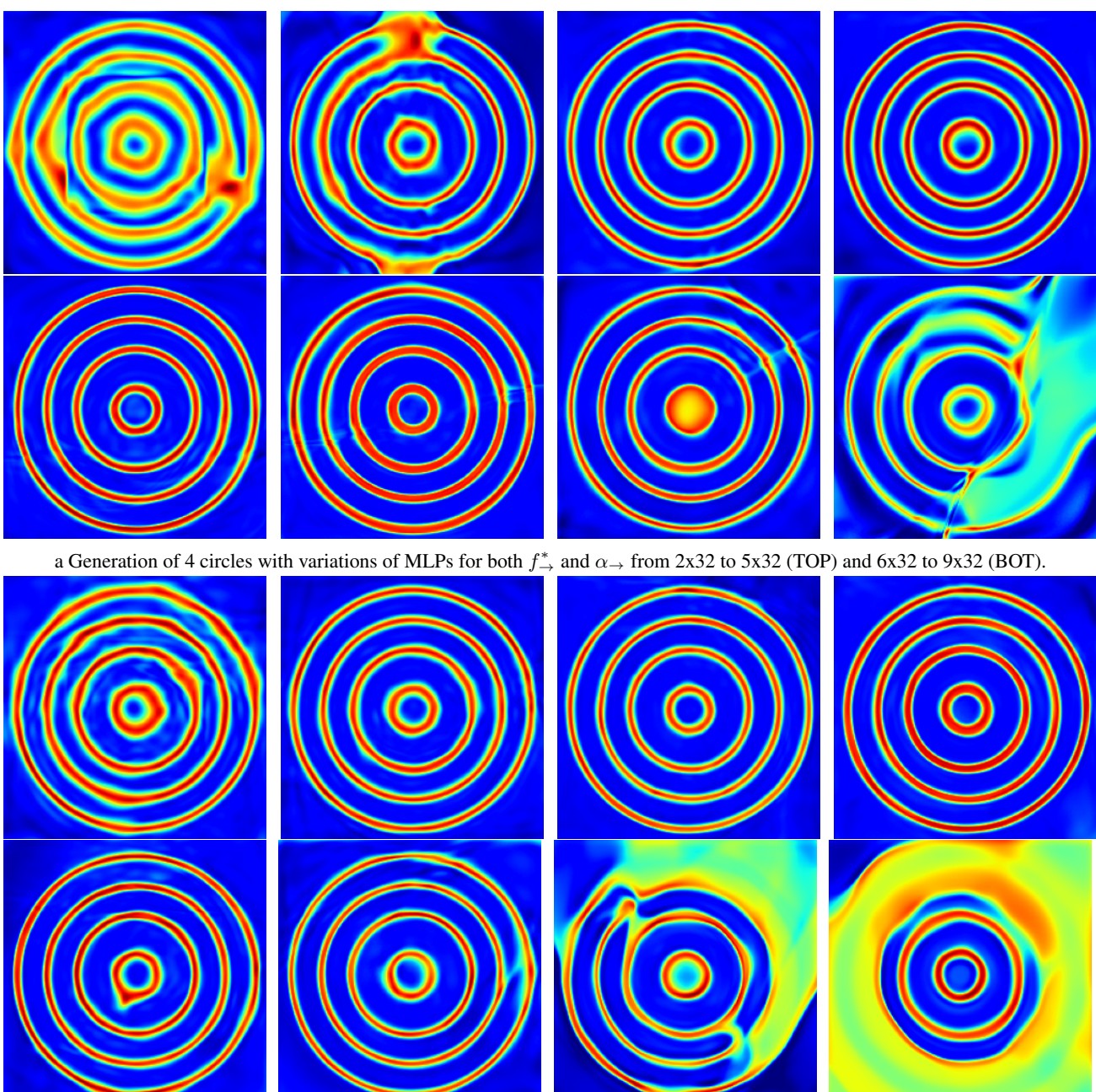

a Generation of 4 circles with variations of MLPs for both $f_\rightarrow^*$ and $\alpha_\rightarrow$ from 2x32 to 5x32 (TOP) and 6x32 to 9x32 (BOT).

b Generation of 4 circles with variations of MLPs for both $f_\rightarrow^*$ and $\alpha_\rightarrow$ from 2x64 to 5x64 (TOP) and 6x64 to 9x64 (BOT).

*Figure 9.* Variation of $f_\rightarrow^*$ and $\alpha_\rightarrow$ depths and width from 2x32 to 9x64. Increasing depth and width increases quality until depth 4 then the training seemingly becomes lest stable as shown by the 9x64 experiment. Since we keep the learning rate constant at 1e-3, it is likely that the MLPs training becomes unstable. The state space is $\mathcal{S} = \mathbb{R}^2$, to ensure $F_\rightarrow(\mathcal{S}) < +\infty$, we constrain $f_\rightarrow^*$ to be zero outside $[-4, 4]^2$. The EGF has 8 transformations which are translations, while the MLPs for $\alpha_\rightarrow$ and $f_\rightarrow^*$ vary from 2x32 (2 hidden layer of width 32) to 9x64. Learning rate is kept at 0.001.

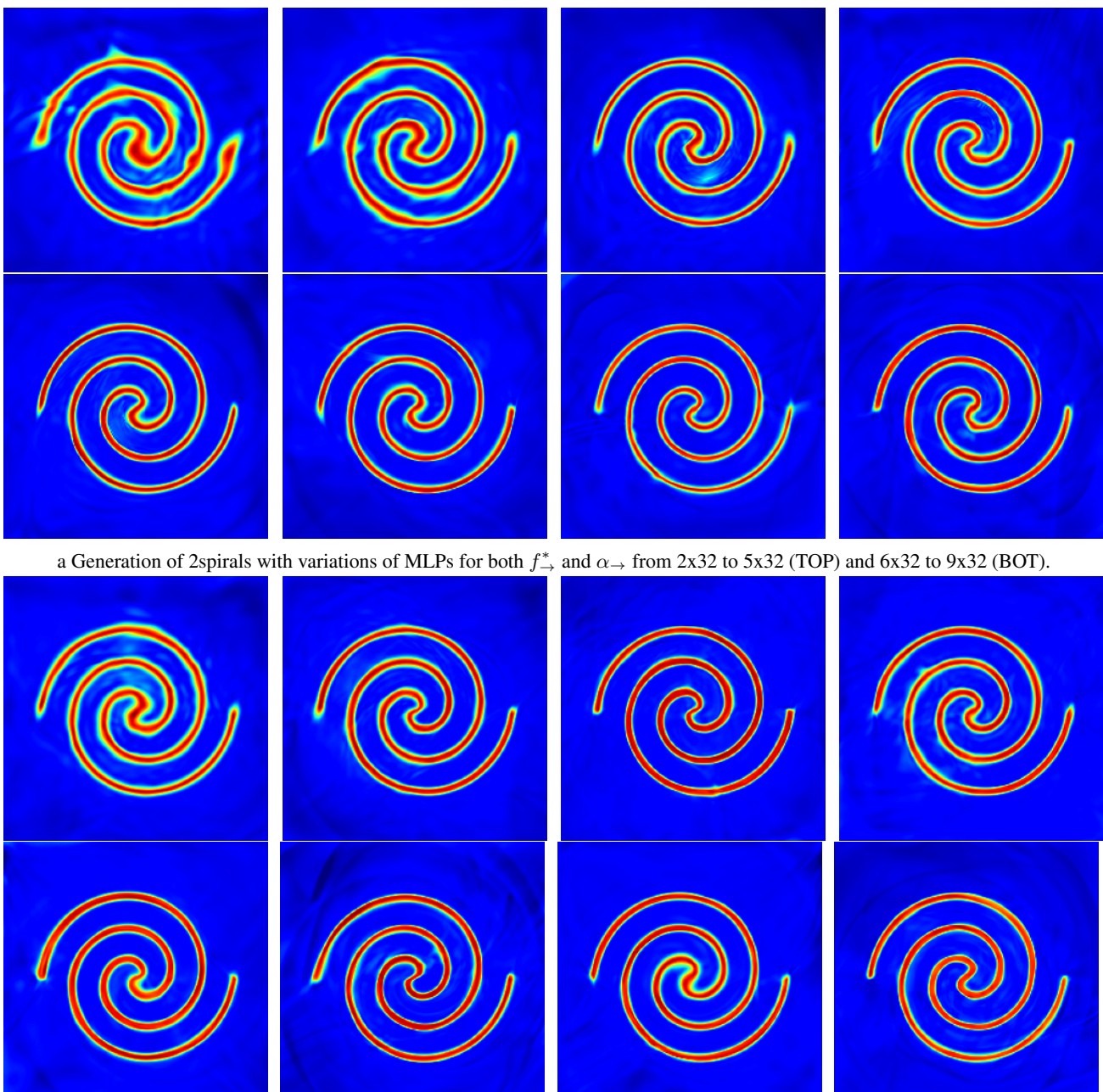

a Generation of 2spirals with variations of MLPs for both $f^*_\rightarrow$ and $\alpha_\rightarrow$ from 2x32 to 5x32 (TOP) and 6x32 to 9x32 (BOT).

b Generation of 2 spirals with variations of MLPs for both $f^*_\rightarrow$ and $\alpha_\rightarrow$ from 2x64 to 5x64 (TOP) and 6x64 to 9x64 (BOT).

*Figure 10.* Variation of $f^*_\rightarrow$ and $\alpha_\rightarrow$ depths and width from 2x32 to 9x64. Increasing depth and width increases quality until depth 4 then the training seemingly becomes lest stable as shown by the 9x64 experiment. Since we keep the learning rate constant at 1e-3, it is likely that the MLPs training becomes unstable. The state space is $\mathcal{S} = \mathbb{R}^2$, to ensure $F_\rightarrow(\mathcal{S}) < +\infty$, we constrain $f^*_\rightarrow$ to be zero outside $[-4, 4]^2$. The EGF has 8 transformations which are translations, while the MLPs for $\alpha_\rightarrow$ and $f^*_\rightarrow$ vary from 2x32 (2 hidden layer of width 32) to 9x64. Learning rate is kept at 0.001.

