# OpenReview forum: "Ergodic Generative Flows"
_ICML.cc/2025/Conference — ICML 2025 poster_

### Official Review · Reviewer_HCTd · 2025-03-11

**Overall Recommendation:** 3

**Summary:**

The paper studies the problem of generative modelling and sampling (a bit confusingly referred to in the paper as IL and RL). The authors focus on the framework on GFlowNets in continuous spaces and learning directly from samples. The paper presents an alternate theoretical framework of Ergodic Generative Flows thats redefines the transition mechanism using a finite set of globally defined diffeomorphisms—smooth, bijective transformations with smooth inverses—that exhibit ergodic properties. Ergodicity ensures that iterative application of these transformations enables comprehensive exploration of the state space, allowing EGFs to approximate any point within it over time. This framing results in the flow matching objective requiring only a discrete sum over the finite set of diffeomorphisms instead of an integral. The paper establishes theoretically that EGFs can approximate any continuous distribution on compact manifolds such as tori and spheres under some assumptions. The paper also introduces KL-weakFM, which integrates a Kullback-Leibler (KL) divergence-based cross-entropy term with a relaxed version of the flow-matching condition, allowing learning directly from samples. The paper also includes some experimental results on low-dimensional problems, with EGFs showing strong performance.

### Update after rebuttal

The authors provided some clarifications to my some of my questions. However, I do not agree with some of the claims made in the rebuttal (about calling what the method is doing imitation learning over generative modeling and TB not being a "pure GFN" and thus not a valid baseline). The authors also did include a baseline with a learned density which is discussed in the paper. Regardless of these, I think the paper makes interesting contributions so I keep my positive rating.

**Claims And Evidence:**

C1: Extending the theory of GFNs with quantitative results about sampling in the non-acyclic case.

To the best of my understanding, the results in TB (Malkin et al. 2022) are applicable with a weaker assumption of termination w.p. 1 instead of acyclicity. Aside from Brunswic et al. 2024 which is discussed briefly in the paper, the result in Theorem 3.8 provides a novel quantitative result for GFNs in non-acyclic domains, providing a bound on the total variation. Although I note that the bound is not tested numerically.

C2: Develop a theory of EGFs which induces a tractable flow matching loss

In Section 3, the authors introduce EGFs which provide an alternate parameterization for policies in terms of a finite set of diffeomorphisms, and establish universality in Theorem 3.4, Theorem 3.5 and Theorem 3.6. The authors also discuss the dependency of the result on the $L^2$ mixing summability property in the Limitations.

C3: Propose a coupled KL-weakFM loss to train EGFs for generative modeling

In Section 3.3, the authors propose and discuss the KL-weakFM loss and validate it numerically in experiments in Section 4 in the generative modeling setting. This is the claim with the weakest evidence as I will elaborate further below. In particular, the experimental choices could be improved significantly.

**Essential References Not Discussed:**

In the context of tractable continuous GFlowNets, Sendera et al. 2024 leverages the view of GFNs are diffusion models to enable scaling to high dimensional problems and would be useful to discuss here.

**Experimental Designs Or Analyses:**

* Additionally I find the baselines in the experiments lacking. For instance, in the RL case there is no comparison with other tractable GFN losses (for continuous spaces) like trajectory balance.
* While the authors talk extensively about the shortcomings of learned diversity, the IL experiments have there is no comparison with a learned density baseline.

**Methods And Evaluation Criteria:**

* A shortcoming already discussed in the paper is that the empirical evaluations are limited to low dimensional problems.

**Other Comments Or Suggestions:**

* (Minor) As the paper is mainly theoretical, the writing can be a bit dense at times. Some rewording, especially in Section 3 to make the results a bit easier to follow would be helpful.
* (Minor) What the authors call imitation learning is really just a problem of generative modelling, and I find calling it IL a bit confusing.

**Other Strengths And Weaknesses:**

Strengths:
* I find the idea of replacing policies with a finite set of diffeomorphisms quite novel and interesting, as it opens up new avenues to explore.

Weaknesses
* (Minor) The paper focuses quite heavily on the intractability of flow matching in continuous spaces and equates it as a weakness of GFlowNets in general, even though there are tractable objectives like TB. I think the framing could be improved in that regards.

**Questions For Authors:**

N/A

**Relation To Broader Scientific Literature:**

The contributions of the paper add to the literature on GFlowNets, In particular the quantitative bound in Theorem 3.8 is useful addition to the broader literature on GFlowNets.

**Theoretical Claims:**

The key contributions of the paper are theoretical in nature. I have read through all the results and looked at the proofs for Theorem 3.4, 3.5, 3.6 and 3.8. I have tried my best to check the correctness but it Is possible that I might have misunderstood some aspects.

---

> ### Author Rebuttal · Authors · 2025-03-27
>
> Dear HCTd,
>
> We thank you for your detailed review. Before going into detail, allow us to emphasize that the present submission is an "Exercise in Style": how far can we go with "pure" GFN without a separate energy model? We understand it may be restrictive, but this is the game we decided to play.
>
>
> 1. You are right to point to Sendera et al., 2024, which deals with GFN generalization of diffusion models. To the best of my knowledge, Sendera et al. are restricted to reward and energy model objectives. As such, we should cite them together with Zhang et al., 2022.
>
> 2. Experimental Designs Or Analyses (1): We agree that DB or TB losses comparison is important, we made the initial choice to restrict ourselves to FM-losses for consistency. Also, the main purpose of the RL section is to check the basics of EGF in a standard GFN setting and confirm the 0-flow problem noted by Brunswic et al. DB or TB losses require changes in our implementation, but we will try to incorporate stability results by the end of the rebuttal discussion.
>
> 3. Experimental Designs Or Analyses (2): We fail to understand the comment as Figure 2 provides such a comparison.
>
> 4. Tractability of TB.   Since TB objective does not enforce flow-matching property (there is no flow, only policies) it's not a "pure" GFN. Furthermore, to the best of our knowledge, TB requires a separate energy model since it has no flow. On the other hand, DB may be in the scope, but the expectancy of the DB loss is the same as the expectancy of the FM loss if the backward policy is chosen according to formulas (6-7), except that DB has a higher variance.
>
> 5. The other reviewers also made several suggestions and remarks regarding rewording. Please see our answers to the other reviews for the intended changes.
>
> 6. We wanted to emphasize the distinction between "target distribution with density" and "target distribution with samples." We felt that generative modeling in AI covers many practices that intersect both worlds (the work of Sendera et al., for instance), and we struggled to find the right terminology. Although imperfect choices, "Imitation Learning" carries the idea of "examples" while "Reinforcement Learning" is strongly associated with "reward" in the community. We cannot change this choice for the present submission, but we are open to suggestions for terminology that you may provide.

---

> > ### Comment · Reviewer_HCTd · 2025-04-03
> >
> > Thanks for the response!
> >
> > > Experimental Designs Or Analyses (2)
> >
> > I think there is a typo in my review it should say learned "density" instead of learned diversity. My apologies! Perhaps I should clarify my question: The discussion in the paper seems to say instead of learning an unnormalized energy and then training a GFN to sample from it, EGFs can learn the sampler directly. So a simple baseline would be learning the _unnormalized_ energy using samples and then training a GFN on it, which as I understand is not the case in Figure 2.
> >
> > > Since TB objective does not enforce flow-matching property (there is no flow, only policies) it's not a "pure" GFN.
> >
> > I am not sure I follow the argument here. The policies learned in TB are defined using the flows, and the flows can be recovered (in principle) using the policies and $Z$. Proposition 1 in the TB paper shows that optimizing the TB objective achieves the flow matching condition. So I am not sure what you mean by "pure" GFN.
> >
> > > We felt that generative modeling in AI covers many practices that intersect both worlds (the work of Sendera et al., for instance)
> >
> > I think "generative models" is typically used to define approximating $p_\theta(x)$ (that can be sampled) given only access to samples $x_i\sim p(x)$. Sendera et al. study the sampling problem where you have access to an unnormalized energy and want to sample from the distribution. So I am not sure why this terminology is a problem.

---

> > > ### Author Response · Authors · 2025-04-03
> > >
> > > 1) As we mentioned in our first reply, our intention was purity, hence the will to train GFN with its flow $F(\cdot \rightarrow s_f)$ as reward model. This was also suggested in GFN foundations https://arxiv.org/pdf/2111.09266 Section 4.5 paragraph 3. However, you are right to point out that we should make the comparison. Our intuition was that beyond purity, the interest only reside two major points and two secondary ones:
> > >     - (Major) the GFN induced reward model is *normalised* by construction if the WeakFM part of the loss is zero. This is a consequence of the equation (17) on page 6. In general, such guarantees are hard to ensure for general proxy reward models and results in more difficult crossentropy training of the reward model. Our main drawback is that we have to deal with a small background reward due to positive flow matching error bias induced by strong WeakFM loss, hence the background filter we use.
> > >
> > >     - (Major) GFN internal reward model come with the guarantee of no mode collapse under zero WeakFM loss. Such a guarantee is more difficult to ensure for external reward models as one has to guarantee full support of the training distribution of the GFN which rely on the policy does not "focusing on modes too quickly" as in adaptive MCMC. See for instance https://link.springer.com/article/10.1007/s11222-008-9110-y . In our case, Ergodicity already ensures full support, so  long trajectories should be sufficient to ensure that the estimator of the WeakFM loss is reliable.
> > >
> > >     - (minor) bagging: having common features for reward on policies. But this may be done in a more straightforward manner by using a common backbone model.
> > >     - (minor) Theorem 3.8 and corollaries are slightly more straightforward without an separate reward model.
> > >
> > > Therefore, our interest is mainly qualitative and theoretical. Nomalization is very easy to check but testing mode collapse  would probably  require to build an a fail case of EBM-GFN with mode collapse. Fairness of such an comparison requires careful experimental design. We cannot promise to do that within the limited time separating us from the end of the current discussion.
> > >
> > > 2) Regarding TB, it may be a personal opinion but TB is to GFN what DPO is to PPO: the flow is implicit.  Worse,  it is intractable in many cases, especially ours, as its computation would require to integrate over all possible trajectories passing through a given point. Yes TB guarantees come from flow reasoning, but this is insufficient to call it a GFN. In the same way, DDPM is a GFN with the density at time $t$ identified with the state flow, but we can agree that sincd only the gradient of the log density is known (ie its mean policy)  diffusion is only a GFN from a Theoretical view point that serves only a pedagogical purpose.  Finally, correct us if we're wrong but, if a GFN was parameterized with a forward policy and a star outflow as we do, the backward policy chosen as we do using a detailed balance condition, then the TB loss would not guarantee flow matching condition in the sense of equation (2) in our work, one would have $\overline F:=\sum_t F_{init}\pi_\rightarrow ^t \neq F_\rightarrow $ on  $\mathcal S$ and possibly $\overline F \otimes \pi_\rightarrow (\cdot\rightarrow s_f)=R\neq F_\rightarrow \otimes \pi_\rightarrow (\cdot\rightarrow s_f)$.
> > > So sampling objective is fulfilled but not the fow matching condition.
> > >
> > > 3) We are quibbling on terminology: EMB-TB training is akin to LLM-SFT. It is definitely generative modeling but the method is two stages and Reward learning based. If we do not do this distinction, the whole "separate energy model" against "internal energy model" discussion becomes more confusing. We may defend our viewpoint choices but we have to agree that our terminology choice was not the best.
> > >
> > >
> > > 4) we have done the NASA flood and earthquake experiments. Here are the NLL scores
> > > Earthquake: -0.12 for EGF  on par with Moser flow -0.09
> > > Flood:  0.56 for EGF on par with Moser flow at 0.62

---

### Official Review · Reviewer_mQ7G · 2025-03-12

**Overall Recommendation:** 4

**Summary:**

This paper introduces Ergodic Generative Flows (EGFs), a novel framework that extends Generative Flow Networks (GFNs) to address key challenges in training generative models for both reinforcement learning (RL) and imitation learning (IL). The authors identify four main challenges with existing GFNs: intractability of flow-matching loss, limited tests of non-acyclic training, the need for a separate reward model in imitation learning, and challenges in continuous settings.
The key innovations of this work are: (1) leveraging ergodicity to build generative flows with finitely many globally defined transformations (diffeomorphisms), providing both universality guarantees and tractable flow-matching loss; (2) introducing a new KL-weakFM loss that couples cross-entropy with weak flow-matching control, enabling IL training without a separate reward model; and (3) developing a mathematical framework for EGFs that ensures expressivity even with simple transformations like translations on tori and rotations on spheres.
The authors empirically validate their approach on toy 2D tasks and real-world datasets from NASA on the sphere using the KL-weakFM loss. Additionally, they conduct toy 2D reinforcement learning experiments with a target reward using the flow-matching (FM) loss. Their results demonstrate that EGFs can effectively address the identified challenges and outperform baseline methods, particularly in settings where computational and time budgets for generation are highly constrained.

**Claims And Evidence:**

Most claims in the paper appear to be well-supported by theoretical analysis and empirical evidence. The authors provide a comprehensive mathematical framework that logically connects EGFs to existing generative modeling approaches.

However, some claims could benefit from additional clarification or support:

- The claim in Theorem 3.4 regarding the expressivity of EGFs lacks specificity about what "expressive enough" means in terms of parameterization of $f^*_{\rightarrow}$. While the theorem states that EGFs can approximate any smooth density, the conditions under which this holds true could be more precisely defined.
- The experimental results in Section 4.1 demonstrate the stability of EGFs with regularization, but the paper doesn't thoroughly explore why regularization has this stabilizing effect—whether it's simply controlling flow size or inducing more meaningful behavioral changes.

**Essential References Not Discussed:**

Besides the Zhang et al. (2022) paper mentioned above, which should be cited given its relevance to extending GFlowNets to imitation learning, I don't have specific knowledge of other missing essential references.

**Experimental Designs Or Analyses:**

The experimental designs in the paper are generally sound. The authors test their approach on both synthetic and real-world data, which helps validate the method's applicability.

In Section 4.1, the authors test EGFs on a checkerboard distribution, which is a good choice for evaluating expressivity. However, the analysis of why regularization stabilizes training could be more thorough.

For the experiment in Section 4.2 on the volcano dataset, the authors appropriately use the KL-weakFM loss from Algorithm 1 and provide comparisons to baselines. The positive results here are compelling evidence for the effectiveness of EGFs.

One potential issue is that Algorithm 1 uses importance weighting terms involving ratios of densities $f^*_{\rightarrow}$ and $f^*_{\leftarrow}$. If these densities are poorly estimated or close to zero in some regions, the weights could become unstable. It would be valuable to know if the authors encountered this issue in their experiments and, if so, how they addressed it.

**Methods And Evaluation Criteria:**

The methods proposed in the paper are appropriate for addressing the challenges in training GFNs. The authors' formulation of EGFs using a finite set of diffeomorphisms with topological ergodicity provides a mathematically sound foundation that enables tractable flow matching while maintaining expressivity.
The introduction of the KL-weakFM loss is particularly noteworthy as it eliminates the need for a separate reward model in imitation learning settings, addressing one of the key challenges identified. By coupling cross-entropy with weak flow-matching control, this approach offers a more direct path to training generative models from demonstration data.
The evaluation criteria used in the experiments are sensible. For the toy 2D tasks, the authors assess the expressivity and stability of EGFs, which are important properties for generative models. For the NASA datasets on the sphere, they use the KL-weakFM loss and evaluate sample quality, demonstrating the real-world applicability of their approach. The additional reinforcement learning experiments with a target reward further validate the flexibility of the framework.

However, the evaluation could be strengthened by:

- Including direct comparisons to standard generative models like diffusion models in computationally constrained settings, which would better highlight the claimed advantages of EGFs.
- Providing more comprehensive metrics such as training time, sampling efficiency, and robustness to hyperparameter choices.
More extensive testing on higher-dimensional problems to demonstrate scalability beyond 2D and spherical domains.

**Other Comments Or Suggestions:**

- The background section should more clearly explain the stopping probability in GFlowNets and how it relates to the terminal distribution. Specifically, it should clarify that when $F_{\text{term}}$ has a lot of mass, the process is likely to stop, and when $F^*_{\rightarrow}$ dominates, it is likely to continue.
- The derivation of Equation 7 should explicitly mention its connection to the detailed balance condition ($\text{Backward flow density} \times \text{Backward transition probability} = \text{Forward flow density} \times \text{Forward transition probability}$) and conservation of probability.
- Definition 3.3 could benefit from a more intuitive explanation, perhaps using the interpretation that it ensures the Markov kernel decorrelates functions over time.
- The authors should explicitly mention that proofs are in the appendix.

**Other Strengths And Weaknesses:**

Strengths:
- The paper presents a compelling unification of different generative modeling approaches, which is valuable for advancing theoretical understanding in the field.
- The introduction tells a coherent and gradual story, building from normalizing flows to diffusion models to GFlowNets, which helps contextualize the work.
- The proposed Algorithm 1 elegantly trains an ergodic generative flow using imitation learning with a balanced loss function.

Weaknesses:
- The presentation of some mathematical concepts is dense and could be more accessible. For example, Equation 7 appears to stem from the detailed balance condition and conservation of probability, but these connections aren't explicitly stated. Most theorems and definitions would benefit from a more gentle introduction for the typical ICML paper reader.
- The background section on GFlowNets could be clearer, particularly in explaining how the stopping probability $\frac{dF_{\text{term}}}{d(F_{\text{term}} + F^*_\rightarrow)}(s_t)$ ensures that the Markov chain naturally stops in regions that match the terminal distribution (in the case of discrete acyclic GFlowNets).
- The paper would benefit from a more thorough comparison to existing generative models in terms of sample quality, training time, and robustness.

**Questions For Authors:**

- In Theorem 3.4, could you clarify what "expressive enough" means in terms of the parameterization of $f^*_{\rightarrow}$? What specific conditions must be met for EGFs to approximate arbitrary smooth densities?
- In Algorithm 1, have you encountered issues with importance weighting terms becoming unstable when densities $f^*_{\rightarrow}$ and $f^*_{\leftarrow}$ are poorly estimated or close to zero? If so, how did you address this challenge?
- Could you elaborate on why regularization stabilizes training in the experiments in Section 4.1? Is it merely controlling the flow size, or does it induce more meaningful changes in model behavior?
- Have you compared EGFs and standard generative models like normalizing flows and diffusion models? How do EGFs compare in terms of sample quality, training time, or robustness?

I'd be happy to raise my score if my questions are addressed and if the authors commit to improving the clarity and readability of their (great) paper, by providing a reasonable plan.

**Relation To Broader Scientific Literature:**

The paper builds upon and extends several strands of research in generative modeling:

- It draws connections to normalizing flows, which model distributions through invertible transformations.
- It relates to diffusion models, which generate samples by reversing a diffusion process.
- It extends the framework of GFlowNets, which were initially designed for generative reinforcement learning.

The paper contributes to unifying these disparate approaches under a common theoretical framework, which is valuable for advancing the field's understanding of generative modeling.

However, the authors should acknowledge that Zhang et al. (2022) in "Unifying generative models with GFlowNets and beyond" already attempted to extend GFlowNets to imitation learning using the Trajectory Balance Consistency objective, achieving positive results. This prior work is relevant to understanding the context of the current paper's contributions.

**Theoretical Claims:**

The paper presents several theoretical results, including:

- Theorem 3.4 on the expressivity of EGFs
- Definition 3.3 of $L^2$-exponential ergodicity
- Theorem 3.8 and Corollary 3.9 on sampling error bounds

The proofs are in the appendix, which should be explicitly mentioned in the main text. While the core ideas behind these results seem sound, Definition 3.3 could be explained more clearly. As I understand it, the concept of $L^2$-exponential ergodicity ensures that the Markov kernel decorrelates functions over time, meaning the influence of initial conditions diminishes rapidly in an $L^2$-sense.

The formulation of Theorem 3.8, which provides bounds on sampling error in terms of total variation distance, is valuable but would be strengthened if expressed for acyclic discrete GFlowNets, for example, using common notation in the field, which would make the connection to existing work more explicit.

---

> ### Author Rebuttal · Authors · 2025-03-27
>
> Dear mQ7G,
>
> We thank you for your particularly detailed review.
> To begin with, we will add clear references to proofs in the appendix.
>
> 3. Theoretical Claims:
> - Your interpretation of Definition 3.3 is correct. We may move the paragraph after Theorem 3.4 to after Definition 3.3 and expand it to better explain this definition by referring to the mixing properties of the Markov kernel.
>
> We agree that understanding Theorem 3.8 would benefit from a specialization on graphs. Your suggestion is in line with our answer to reviewer 6zNQ regarding neural network approximations. We shall insert a paragraph answering both points in the RL setting.
>
>
> 5. Scientific Literature:
>
>  - We apologize for not citing Zhang et al.  We will cite it properly.
>
> 6. Strengths And Weaknesses:
>
> - Equation 7, you are right; this equation should be introduced more gently. It may be derived from
> $\pi^*_\leftarrow (x\rightarrow A) = \frac{d (F_{\rightarrow}^*\otimes \pi^*_\rightarrow)(A\rightarrow \cdot)}{d F_{\leftarrow}^*}(x)$
> taken from section 3.2 of Brunswic et al. https://arxiv.org/pdf/2312.15246. This equation indeed formalizes a detailed balance condition in the measurable setting. Also, Proposition 1 in the appendix clarifies the link between bi-measures and Markov kernels. We will add a sentence to account for this interpretation and add proof in the appendix.  More generally, we will enhance the introduction of Theorems of definitions with comments on their intuition.
>
> 7. Questions:
>
> - The main point of Theorem 3.4 is that EGFs reduce the generative problem of finding a scalar field on the state space. Theorems 3.5 and 3.6 provide sufficient conditions for L2-exponential mixing, then an L2 error on $f_{\rightarrow}^*$ translates into an increase in  $\delta$ in Corollary 3.9 and an increase of the TV, which is easy to relate to the L1 error of $f_{\rightarrow}^*$.
> We agree that the "expressive enough" formulation is vague and requires a comment.
>
> - Regarding Algorithm 1 and importance densities. The densities were used as is for the results presented in the submission.  In later experiments, we added a small constant (1e-2) to stabilize training; however, no ablation studies were conducted to assess its relevance or effectiveness. Flow collapse was the most common problem encountered but seemed more related to too strong a regularization than to collapsing densities. A lower bound on $\int f_{init}$ proved sufficient to force exploration.
>
> - Your comment on the regularization echoes that of moVi. The regularization is inspired by Brunswic et al. regularization; their Theorem states that a decaying regularization may be used to enforce convergence toward an acyclic flow on graphs when using a stable FM loss. From an optimization viewpoint, the regularization pushes the derivative of the loss to be positive in the direction of any 0-flow, hence leading to a stable loss in the sense of definition 3 of Brunswic et al.  In our setting, any 0-flow component is expected to vanish (but the theorem mentioned above does not apply to our setting due to their enumerable state space assumption). A side effect is reducing the sampling length.  We will enhance the RL section with a discussion of this regularization.
>
>
> - We conducted comparisons to diffusion models (DDM) and normalizing flows (FFJORD), but we felt that the comparison to Moser flow already carried our point. We will add qualitative comparisons in the appendix on 2D toys on tiny 32x2 models. On the NASA datasets, we already include NLL comparisons to baselines for volcanoes; we will add the results on earthquakes and floods, also the results for Riemannian diffusion, as well as a comment on the relative size of each baseline.
>     - The experiments presented in the paper were solved with manual hyperparameter tuning; the only caveats were:
>         - too high regularization scale leads to flow collapse
>         - too high WeakFM scale tends to make learning the optimum harder especially for small models.
>
> So far, we have only trained EGFs up to dimension 20 on distributions (toy and sparse reward Markov decision tasks). However, we prefer to limit the scope of the present work to proof of concept as hyperparameter tuning proved less straightforward in higher dimensions.
>     - Regarding training time,  we did not benchmark the training speed. We shall include a comparison table in the appendix.  Our wall clock evaluation led us to conclude that our training for NASA datasets was faster than that of Moser Flow by several orders of magnitude. Diffusion models tend to train slightly faster than EGFs on 2D toys, while FFJORD is significantly slower. However, hyperparameters have been shown to impact convergence speed significantly.

---

> > ### Comment · Reviewer_mQ7G · 2025-04-04
> >
> > Thank you for your detailed reply and your eagerness to improve the clarity of the paper.
> >
> > Upon reading your answer, and other reviews, I am raising my score.

---

> > > ### Author Response · Authors · 2025-04-04
> > >
> > > We thank you.

---

### Official Review · Reviewer_6zNQ · 2025-03-12

**Overall Recommendation:** 3

**Summary:**

This work introduces Ergodic Generative Flows (EGFs) to tackle several issues not satisfactorily resolved in generative flow networks by using ergodicity to create simple generative flows with globally defined transformations and tractable flow-matching loss. Furthermore, a new KL-weakFM loss is proposed for IL training without a separate reward model. The effectiveness of IL-EGFs is evaluated on toy 2D tasks and real-world NASA datasets, while toy 2D reinforcement learning experiments are conducted using the FM loss.

**Claims And Evidence:**

This paper is well-written and presents rigorous theoretical results for a class of generative models. The authors provide clear and convincing evidence to support their findings.

**Essential References Not Discussed:**

It appears that the essential related works are adequately cited in this paper.

**Experimental Designs Or Analyses:**

The experiments conducted to assess the effectiveness of the proposed method are limited to two-dimensional datasets. While these experiments provide initial insights into the method's performance, evaluating it on higher-dimensional datasets would offer a better understanding of the method.

**Methods And Evaluation Criteria:**

This paper primarily focuses on theory and methodology, offering substantial theoretical insights and proposing new methods. However, the experimental evaluation of the proposed method is somewhat limited, with only a few experiments conducted to assess its effectiveness. Expanding the experimental section with more extensive tests could further validate the practical applicability and performance of the method.

**Other Comments Or Suggestions:**

No other comments.

**Other Strengths And Weaknesses:**

No additional comments here.

**Questions For Authors:**

I have the following questions for the authors:

1. In the standard generative learning scenario, where only a random sample of size n is available, do Theorem 3.8 and Corollary 3.9 provide any insights into how the error bounds are influenced by the sample size?

2. How do the error bounds in Theorem 3.8 and Corollary 3.9 relate to the dimensionality of the data? Do these results suffer from the curse of dimensionality?

3. Which functions are approximated by neural networks in your framework, and how are the approximation errors addressed in Theorem 3.8?

**Relation To Broader Scientific Literature:**

Generative Flow Networks (GFNs) were initially developed for sampling from unnormalized distribution densities on directed non-acyclic graphs. While recent advancements have expanded their theoretical framework, challenges persist in training GFNs in continuous settings and for imitation learning (IL), such as intractable flow-matching loss and the need for a separate reward model. The key contributions of this paper is that it addresses these issues and proposed a new approach.

**Theoretical Claims:**

I have gone over the theoretical results and proofs presented in the paper, and they appear to be correct. The logical flow and mathematical rigor of the arguments are sound, and the conclusions drawn from the proofs are well-supported.

---

> ### Author Rebuttal · Authors · 2025-03-26
>
> Dear  6zNQ,
>
>    We thank you for your detailed review.
> We acknowledge the need for higher dimensional experiments, it is part of an ongoing project to scaling up EGFs as well as training conditioned EGFs.
>
> Regarding your questions:
> 1. Unfortunately, they do not directly.  A straightforward strategy would be to use Wasserstein bounds together with dequantization.
>     - By Theorem 3.8 and its corollaries, the KL-WeakFM loss controls the TV distance between the EGF sampling distribution and the target dequantized empirical distribution.
>     -  On compact domains W < diameter * TV, the loss then controls the Wasserstein distance between EGF sampling distribution and dequantized empirical target distribution.
>     - Finally, use triangular inequality in W distance for the path "dequantized empirical target distribution -> empirical target distribution -> true target distribution."  Bounds such as https://arxiv.org/pdf/1707.00087 to estimate the Wasserstein distance of the empirical distance to the target distribution are leveraged at this step.
>
>     Such a computation is straightforward and may be added in appendix. The unbounded case requires extra care and assumption for the second step.
>
> 2. Regarding the curse of dimensionality,
>     - to begin with, we refer to the answer given to reviewer moVi about the control of the sampling time (short answer yes in a tame way).
>     - more to the point, the curse of dimensionality is likely to arise in the estimation of the weakFM loss since we need to enforce a flow-matching condition. We expect the spectral gap to reduce the asymptotic theoretical complexity of the Monte Carlo estimator compared to naive translation moves (non-ergodic), but more theoretical work is needed. A more sophisticated replay buffer is also probably required.  We note this question for future theoretical investigations.
>
> 3. On neural networks approximations
> - Neural networks approximate:
>     - the star outflow $f_\rightarrow^*$,
>     - the policy softmax part $\alpha_\rightarrow^i$.
>     - the translation part of the affine transforms on tori (but with no inputs since they do not depend on the state)
>
> - The answer to your question then depends on whether we are in the RL or IL setting. In the IL setting, Corollary 3.9, together with equation 16, ensures that the KL-WeakFM loss controls the TV error. In the RL setting, we should add a paragraph explaining the following:
>     - an L1 flow matching error controls the TV bound, itself bounded the L2 flow matching error controlled by the loss, which in turn depends (in a tractable way) on the trainable models $f_\rightarrow^*$, $\alpha_\rightarrow^i$ and the fixed models $f_{init},f_{term}$. More abstractly, the FM loss we use is an estimator of a twisted L2 distance of the density of the approximated flow bimeasure $F_\rightarrow \otimes \pi_{\rightarrow}$  to the affine subspace of target flow-matching bimeasures, where $F_\rightarrow = (f_\rightarrow^* + f_{term})\lambda + f_{init}\delta_{s_0}$ and $\pi_{\rightarrow}(x) =  \frac{  f_\rightarrow^*(x)}{ f_{term}(x) + f_\rightarrow^*(x)} \pi_{\rightarrow}^*(x) + \frac{ f_{term}(x)}{ f_{term}(x) + f_\rightarrow^*(x)}\delta_{s_f}$. This L2 distance controls the L1 distance if we assume the state space to have a finite volume. The L1 distance is $\delta F_{init} (\mathcal S)$.
> As a result, the stable FM-loss controls the right-hand side of Theorem  3.8.
>
> Thus, we will add our answer to question 1 in the appendix and a paragraph in the main text to answer question 3.

---

### Official Review · Reviewer_moVi · 2025-03-18

**Overall Recommendation:** 3

**Summary:**

This paper proposes a new family of generative flows called Ergodic Generative Flows (EGFs) which are capable for both RL and IL tasks in continuous settings. The generative flows are built upon finitely many globally defined transformations, with probable universality over continuous spaces like tori and spheres, enabling explicitly expressing and sampling from the undertraining distribution simultaneously. Further, the authors derived a novel loss, coined KL-weakFM loss for IL training where the target distributions is intractable. The proposed methods are evaluated through toy 2D experiments for both RL and IL tasks, and NASA volcano dataset for IL task.

**Claims And Evidence:**

Yes

**Essential References Not Discussed:**

No.

**Experimental Designs Or Analyses:**

Section 4.1 (RL Experiments) is confusing. Despite the issues mentioned above, the purpose of this experiment seems unrelated to the primal claims of this paper, and the presentation of the results (Figure 1(b)) is unclear.

The experiments and analyses in Section 4.2 are reasonable, but I suggest the authors to present the complete results on the complete earth and climate datasets, including volcano, earthquake, flood and fire.

**Methods And Evaluation Criteria:**

The training losses for RL tasks seem problematic. The "stable FM loss" referred to in Section 2 of this paper is neither the commonly used one proposed by Bengio et al. (2021) nor the one proposed by Brunswic et al. (2024). Also, the regularization term in Section 4.1 needs further explanation.

**Other Comments Or Suggestions:**

Some notations are inconsistent or used without definition. For example $f^*_\rightarrow$ in (3), $\delta$ in (5), source and target distribution in Algorithm 1.

**Other Strengths And Weaknesses:**

Strengths
This paper is well-motivated and easy to follow. The proposed method seems to have a strong theoretical support and is insightful. The novel framework of GFlowNet for continuous settings and IL tasks has great potential.
Weaknesses
The RL part lowers the overall level of this paper.
The experiments are limited.

**Questions For Authors:**

Can the authors explain how the sampling time $\tau$ is related to the trainable parameters, the dimension of the state space and the performance of the flow network?

**Relation To Broader Scientific Literature:**

This paper extends the theoretical framework of GFlowNet for continuous settings and IL taks, providing a novel approach for generative models in continuous space.

**Theoretical Claims:**

I checked the theoretical claims except for Section 3.1 because I'm not familiar with topology and differential geometry.

---

> ### Author Rebuttal · Authors · 2025-03-26
>
> Dear moVi,
>
> We thank you for your detailed review and valuable feedback. Please find our responses to your comments below:
>
> 1. Methods and Evaluation Criteria
> Stable Loss:
> We appreciate your observation regarding the difference between our stable loss and the one proposed by Brunswic et al. (2024) in equation (19). We would like to clarify that Brunswic et al. introduce a family of stable losses in Section 4.2, denoted by $\Delta_{f,g,\nu}(\alpha, \beta)$. Our stable loss, as given in Eq. (3), is an instance of this family. Since our focus in this work is on the foundational aspects of Ergodic Generative Flows (EGF), we opted for the simplest stable loss, as it seemed more relevant for our goals. Furthermore, we experimented with Brunswic et al.'s loss on non-acyclic GFlowNets and observed little improvement, with additional difficulties in hyperparameter tuning.
>
> Regularization:
> Brunswic et al. introduced a cycle-killing regularization schedule in Theorem 1 to ensure convergence of non-acyclic flows toward an acyclic flow. Although they do not explicitly suggest using this regularization to stabilize losses in non-acyclic settings, it seemed a natural step for us. We note that a similar approach was also taken by Morosov et al. in their work "Revisiting Non-Acyclic GFlowNets in Discrete Environments."
>
> While we believe the first point does not require further discussion in the main paper, we will include a clarification about our architectural and hyperparameter choices in the appendix. We will also ensure the regularization approach is more clearly explained in the main text. Also there was a typo on the Bengio loss (the square was missing).
>
> 2. Experimental Designs or Analyses
> Purpose of RL Experiment:
> The RL experiment serves a dual purpose: A) testing the applicability of EGF in the original context of GFlowNets, and B) addressing limitation 4. While limitation 4 is secondary to the primary discussion on non-acyclic GFlowNets, we think it is an interesting point that warrants clarification.
>
> Additional Datasets:
> We agree that the inclusion of flood and earthquake results would improve the completeness of our experiments. We are currently running updated experiments to incorporate these benchmarks. However, we were unable to locate the original fire dataset due to changes in the NASA repository.
>
> 3. Supplementary Material
> We have corrected the issues in the supplementary material as per your suggestions.
>
> 4. Other Comments or Suggestions
> We will make the necessary corrections and reformulations to the notations and definitions as suggested.
>
> 5. Sampling Time and Performance
> - Spectral Gap and Sampling Time:
> The estimation of sampling time can be derived from the spectral gap and the $L_2$ norm of the target distribution. We intend to discuss these relations in a broader follow-up work focused on spectral gap control for EGF. Although this theoretical framework is still in development, the following informal results provide some insight:
>
>     - A spectral gap $\eta>0$ guarantees exponential decay in $L^2$ deviation from the mean: $||(f-\int f)\cdot(\pi^*_\rightarrow)^t||  = O((1-\eta)^t).$
>
>     - To transition from an initial distribution $f_{init}$ to a terminal distribution $f_{term}$ with error $\epsilon$, we require: $||(f_{init}-f_{term})(\pi^*_\rightarrow)^t||\leq \epsilon$.
>     - Thus, the required sampling time $\tau$ is expected to scale as: $\tau = O\left(\frac{\log(\epsilon)-\log(||f_{init}-f_{term}|| )}{\log(1-\eta)}\right)$.
>     - One could then leverage general lower bounds on spectral gaps such as those of https://arxiv.org/abs/2306.12358 to guarantee a spectral gap scaling as $\eta \sim 1/dim$.
>     - Furthermore, the difficulty of the objective is controlled by $||f_{init} - f_{term}||$, the co-dimension $q$ of the manifold supporting the distribution would then leads (with a bit of gaussian dequantization) to $\log||f_{init} - f_{term}|| \sim \log Vol($Ball of radius $r$ in dim $q) $ so that one would end up with $\tau = O\left(-\log(\epsilon)dim+q\log(q)dim\right)$.
>     -  However, this bound is probably very loose: in the small volume limit, the "effective" spectral gap is bounded away from 0 (we may discuss this in more details if you wish) so the term $q\log(q)dim$ becomes $q\log(q)$.
>     - We stress the need for a proper analysis, hence a later submission.
> - Trainable Parameters and Sampling Time: the simplest answer is via Theorem 1 as the sampling time is directly controlled by $\int  f^*_{\rightarrow}$ and the WeakFM loss via equation 17.
> The master Theorem 3.4 provides a control:  The proof controls the $L^2$ norm of the target $\||f^*_{\rightarrow}-\int f^*_{\rightarrow} \|| = O(\||f_{init}-f_{term}\||\sum_t(1-\eta)^t) = O(\||f_{init}-f_{term}\||\frac{1}{1-\eta})$. One could link some measure of expressivity  of neural network to such a bound, we did not do it at this stage as we would like to reserve it for a future work (as mentioned above).

---

### Decision · Program_Chairs · 2025-05-01

**Decision:**

Accept (poster)

**Comment:**

This paper proposes a novel view on GFlowNets / hierarchical generative models in continuous state spaces. Motivated by some limitations of continuous parametric kernels (as, for example, in the GFN training algorithms for diffusion samplers), it is proposed to model the transitions by a finite collection of trainable diffeomorphisms of the underlying space and to train policies that select which of them to use. Under an ergodicity condition on the diffeomorphisms, this allows to sample any well-behaved distribution in the limit. Losses are described and tested in both the distribution-matching and data likelihood maximisation settings.

Overall, reviewers found the idea elegant, and I concur. In the responses, the authors committed to improving the clarity of the writing so as to make the main text less dense. The experiments were found borderline-sufficient, and the authors are encouraged to add discussion and comparison with more standard settings where GFNs have been used (e.g., diffusion samplers).

Note: Is there an error in the first sentence of the abstract: GFNs were initially introduced on **acyclic** settings?